



# Assessment of the ISBA Land Surface Model soil hydrology using four closed-form soilwater relationships and several lysimeters

Antoine Sobaga[1,2], Bertrand Decharme[2], Florence Habets[1], Christine Delire[2], Noële Enjelvin[3], Paul-Olivier Redon[4], Pierre Faure-Catteloin[5], and Patrick Le Moigne[2]

[1]Laboratoire de Géologie - CNRS UMR 8538 - École Normale Supérieure - PSL University, IPSL, Paris, France
[2]Centre National de Recherches Météorologiques, Université de Toulouse, Météo-France, CNRS UMR 3589, Toulouse, France
[3]Laboratoire Sols et Environnement-GISFI, Université de Lorraine (UMR 1120), Vandœuvre-lès-Nancy, France
[4]Andra, Direction RD, Centre de Meuse/Haute-Marne, 55290 Bure, France
[5]Université de Lorraine, CNRS, LIEC, F-54000 Nancy, France

**Correspondence:** Antoine Sobaga (sobaga@geologie.ens.fr)

**Abstract.** Soil water drainage is the main source of groundwater recharge and river flow. It is therefore a key process for water resource management. In this study, we evaluate the soil hydrology and the soil water drainage, simulated by the Interaction-Soil-Biosphere-Atmosphere (ISBA) land surface model currently used for hydrological applications from the watershed scale to the global scale. This evaluation is done using seven lysimeters from two long term model approach sites measuring hourly
water dynamics between 2009 and 2019 in northeastern France. These 2-meter depth lysimeters are filled with different soil types and are either maintained bare soil or covered with vegetation. Four closed-form equations describing soil water retention and hydraulic conductivity functions, are tested: the commonly used equations from Brooks and Corey (1966) and van Genuchten (1980), a combination of the van Genuchten soil water retention function with the Brooks and Corey unsaturated hydraulic conductivity function, and, for the very first time in a Land Surface Model (LSM), a modified version of Van
Genuchten equations, with a new hydraulic conductivity curve proposed by Iden et al. (2015). The results indicate a good performance by ISBA with the different closure equations in terms of soil volumetric water content and water mass. The drained flow at the bottom of the lysimeter is well simulated using Brooks and Corey (1966) while some weaknesses appear with van Genuchten (1980) due to the abrupt shape near saturation of its hydraulic conductivity function. The mixed form or the new van Genuchten hydraulic conductivity function from Iden et al. (2015) allows solving this problem and even improves the
simulation of the drainage dynamic, especially for intense drainage events. The study also highlights the importance of the vertical heterogeneity of the soil hydrodynamic parameters to correctly simulate the drainage dynamic, as well as the primary influence of the parameters characterizing the shape of the soil water retention function.

## 1 Introduction

Drainage water is the portion of precipitation that flows through the first meters of soil. As it has escaped evapotranspiration, it
is the main diffuse source of groundwater recharge (Moeck et al., 2020), and is of crucial importance for estimating the evolution of the aquifer (Bredehoeft, 2002; Döll and Fiedler, 2008). Even where there is no aquifer, this water flux can contribute to





the baseflow. Despite their importance, direct observations of drainage water are still rare compared to direct observations of river flow or groundwater level.

Indirect methods based on analysis of baseflow (Meyboom, 1961), or variation of piezometric level for instance using a water table fluctuation method (Healy and Cook, 2002) can be applied to estimate groundwater recharge. However, these methods cannot separate the different components of the recharge, such as the exchange between surface water and groundwater (Brunner et al., 2017; Keshavarzi et al., 2016), exchanges between several aquifer layers basins (Tavakoly et al., 2019), and drainage water. Direct measurement at the local scale and high frequency can be made in situ using lysimeter columns, which isolate a

small volume of soil from lateral flow and collect drainage water. Most of the time, the lysimeters are disconnected from the soil and avoid the capillary rise from the deeper soil that can have an important influence locally (Vergnes et al., 2014; Maxwell and Condon, 2016). From such observations, drainage water is known to have large variations in space, due to changes in soil texture and structure (Vereecken et al., 2019; Moeck et al., 2020).

Due to the limited number of observations and the difficulty to indirectly quantify it, the estimation of the groundwater recharge is one of the twenty-three unsolved problems in hydrology (Blöschl et al., 2019). The simulation of recharge in hydrological models can vary significantly, from simplified reservoir approaches to physically based models. The widely used reservoir approach (Alcamo et al., 2003; Harbaugh, 2005) has the advantage of limiting the number of parameters to be calibrated and reducing the numerical cost of simulations. Physically based approaches are more complex and are commonly

used in LSMs in which vertical water and energy balances between the land surface and the atmosphere can be calculated (De Rosnay et al., 2003; Blyth et al., 2010; Boone et al., 2000).

Today, LSMs are widely used in hydrological applications in order to study the regional and global water cycle, predict streamflow, and inform water resource management (Haddeland et al., 2011; Vincendon et al., 2016; Schellekens et al., 2017;

Gelati et al., 2018; Le Moigne et al., 2020; Vergnes et al., 2020; Muñoz Sabater et al., 2021; Rummler et al., 2022). However, LSMs may have difficulty estimating the dynamics of groundwater recharge, particularly during intense precipitation events. Vereecken et al. (2019) suggested a number of directions for improvement: introduce more physical processes such as preferential flow along the roots and macropores, improve the representation of soil/vegetation and of soil parameters, and improve the spatio-temporal distribution of precipitation. In LSMs, the Richards equation (Richards, 1931) is used to describe the flow

of water in the porous soil due to the actions of capillarity and gravity. Mainly used in the hydrology community, this equation has been widely criticized, in particular for the overestimation of the effect of capillarity (Nimmo, 2010; Farthing and Ogden, 2017).

In the Richards equation, two closed-form equations are often used to represent the variations of volumetric water content

with the matrix potential and the hydraulic conductivity in the unsaturated zone: Brooks and Corey (1966) and van Genuchten (1980), hereafter BC66 and VG80 respectively. Historically, the closed-form equations from BC66 are mostly used by the





atmospheric community in theirs LSMs, while the ones from VG80 are mainly used by hydrologists. BC66 proposed simple analytical power curves of soil water retention and hydraulic conductivity based on North American soil observations. However, they do not include an inflection point close to saturation, and are thus not derivable, which leads to problems at the

connection with the saturated zone. The consequence is a deviation near saturation of volumetric water content. VG80 proposed an improvement of the BC66 soil water retention curve close to water saturation, using more complex analytical forms based on European soil observations. However, as summarised by Iden et al. (2015), when such S-shaped function is used to described fine-textured soil or a soil with a wide pore-size distribution, the VG80 hydraulic conductivity curve exhibits an abrupt drop at the transition from saturated conditions to unsaturated conditions. This may lead to underestimate the hydraulic

conductivity for very wet conditions, i.e. near saturation, and impact the stability and the accuracy of the numerical solver (Vogel et al., 2000; Ippisch et al., 2006). The non linear form also increases the cost of the numerical solvers (van Genuchten, 1980; Vogel et al., 2000; Ippisch et al., 2006; Dourado Neto et al., 2011).

To smooth artifacts associated to the VG80 hydraulic conductivity curve, some studies suggest different solutions: to shift

the entire pore-size distribution by an air-entry value (Kosugi, 1994), to truncate the pore-size distribution by introducing a maximum pore radius on the VG80 hydraulic conductivity curve (Schaap and Van Genuchten, 2006; Iden et al., 2015), or to put an explicit air-entry pressure in the VG80 model (Fuentes et al., 1992; Braud et al., 1995; Vogel et al., 2000; Ippisch et al., 2006). Fuentes et al. (1992) or Braud et al. (1995), proposed to combine the VG80 soil water retention function with the BC66 hydraulic conductivity curve. Such combination should improve the simulation when the soil water is close to saturation, while

preserving the simplicity and numerical stability of the BC66 relationships. It benefits from the many methods that link soil parameters from one relationship to another (van Genuchten, 1980; Lenhard et al., 1989). Recently, an elegant parametrization was proposed by Iden et al. (2015) which allows to keep the VG80 hydraulic conductivity curve by introducing the concept of maximum pore radius suggested by Durner (1994).

The goal of this study is to use a state-to-the-art LSM to assess the benefits of BC66 and VG80 relationships and two of its derivative in reproducing soil water mass, volumetric water content and drainage flux observed in seven lysimeters during more than five years. The two alternative approaches are i) the one proposed by Fuentes et al. (1992) and Braud et al. (1995) that combines the VG80 soil water retention curve with the BC66 unsaturated hydraulic conductivity curve and ii) the new parameterization of the VG80 hydraulic conductivity curve from Iden et al. (2015) that has never been used in a LSM to the

best of our knowledge. The soil hydrodynamic parameters are directly derived from observation and compared with several pedotransfer functions (hereafter PTF) commonly used by LSMs. The LSM used is the multi-layer diffusion version of the Interaction-Soil-Biosphere-Atmosphere (ISBA) that solves directly the mixed form of the Richards equation (Boone et al., 2000; Decharme et al., 2011). The experimental protocol, including descriptions of both the lysimeters data and the ISBA model, is given in section 2. Soil parameters estimation based on lysimeters observations is described in section 3, while the

main results of each model approach are presented in section 4. Finally, a general discussion and the main conclusions are given section 5.



## 2 Experimental Protocol

### 2.1 Data

Seven lysimeters of two French experimental sites located in north-eastern France are used in this study: 4 lysimeters from
the GISFI experimental station (French Scientific Interest Group on Industrial Wastelands) at Homécourt (49°21'N, 5°99'E,
altitude 430m) with data record from 2009 to 2016 (hereafter G1, G2, G3, G4) and 3 lysimeters from OPE experimental station
(Perennial Environmental Observatory) close to Osne-le-Val (48°5092'N, 5°2119E, altitude 224m) with data record from 2014
to 2019 (hereafter O1, O2, O3). These two sites are separated by a distance of 97 km. Their soils are classified according to
the World reference base for soil resources WRB (Hazelton and Murphy, 2016).


The GISFI experimental station focuses on the understanding of pollution evolution and the development of decontamination
solutions (Lemaire et al., 2019; Huot et al., 2015; Rees et al., 2020; Ouvrard et al., 2011). It participates in the collaborative
study TEMPOL (Observation sur le long TErMe de sols POLlués) with the German observatory infrastructure TERENO (Ter-
restrial Environmental Observatories, Zacharias et al. (2011)) in order to study in situ the transfer of pollutants (Leyval et al.,
2018). Three of the GISFI lysimeters (G1, G2 and G3) contain rebuilt soil of the Spolic Toxic Technosol which was sampled
in Neuves-Maisons, an industrial wasteland of a coking plant with contamination (PAH, hydrocarbons, Monserie et al. (2009);
Ouvrard et al. (2011); Sterckeman et al. (2000)). These three lysimeters were filled in September 2007 and soil material was
gradually and manually compacted every 0.3 m to reach a given bulk density. Lysimeters G1 and G2 were maintained bare soil,
while lysimeter G3 was covered by vegetation (Alfalfa). Lysimeter G4 was filled by a monolith of Cambisol from Noyelles-
Godault and covered by grass (Table 1).

The main objective of the OPE site is to describe the environment before and after the construction of the surface facilities
of a deep radioactive waste repository and to follow its evolution. This site is part of the OZCAR (Observatoires de la Zone
Critique: Application et Recherche) French research infrastructure dedicated to the observation and study of the critical zone
(Gaillardet et al., 2018; Braud et al., 2020). The lysimeters were filled by monoliths taken closed to the OPE site: lysime-
ters O1 and O3 were filled with Hypereutric Cambisol with different layers of limestone more or less cracked and lysimeter
O2 contains a Cambisol. These three lysimeters had sparse vegetation, composed of C3-type grass (Table 1). Monolith-filled
lysimeters preserve original soil horizons and are therefore better representative of watershed conditions.

All the lysimeters are weighable cylinders with a depth of 2 m and an area of 1 m$^2$. No suction are imposed at the bottom of
each lysimeters contrary to the ones of TERENO-SoilCan (Putz et al., 2016). They are equipped with suction and temperature
probes as well as time-domain reflectometry (TDR) probes to measure the water content, at 0.50, 1.0 and 1.5 m depth, with
additional measurement at 0.2 m for the OPE lysimeters. The weight is measured continuously with a 0.1 kg precision giving
an idea of the time variations of the total soil water volume in the column, and a tipping counter measures the drainage water
at the bottom. Data are continuously monitored on an hourly basis using a data logger. On the GISFI site, TDR probes are



RIME-PICO32 sensors with internal TDR-electronics. They are set horizontally and record the water content in $\mathrm{cm}^3.\mathrm{cm}^{-3}$ on an hourly basis. The calibration is performed on two measurements, one in dry and one in water-saturated condition. At the OPE site, the soil moisture sensors used (UMP-1Umwelt Geräte Technik GmbH) are based on a frequency domain reflectometry (FDR) method and measure local change in dielectric permittivity. Tipping bucket resolution is $0.1\ \mathrm{mm.h}^{-1}$ on the two sites.


The sets of atmospheric forcing variables (wind speed, precipitation rate, short-wave incident radiation, air temperature, air humidity, atmospheric pressure) used to force the ISBA LSM are observed in situ at an hourly time step by two local meteorological stations, one at OPE and one at GISFI. The atmospheric forcing is assumed to be identical for all the lysimeters of each site. Annual precipitation is 20 % higher at the OPE site (876 $\mathrm{mm.year}^{-1}$) than at the GISFI experimental station (727

$\mathrm{mm.year}^{-1}$). Long-wave radiations are derived from atmospheric conditions using the equation of Prata (1996). Atmospheric data gaps are filled by regressions on available data using two neighboring meteorologic stations. The gaps represent up to 12 % of the observations for the GISFI site.

All soils for all lysimeters are described in Table 1 while all available observations and their characteristics are summarized
in Supplement Table S2.

## 2.2 ISBA Model

The ISBA LSM was originally scheduled for use in numerical weather prediction and climate models. These last decades, ISBA has evolved from a simple bucket force-restore model (Noilhan and Planton, 1989) to a more explicit multi-layer diffusion scheme (Boone et al., 2000; Decharme et al., 2011). ISBA is currently used in hydrological applications, especially
to estimate groundwater recharge when it is associated with hydrologic models at both the regional (Le Moigne et al., 2020; Tavakoly et al., 2019; Vergnes et al., 2020) and the global scales (Vergnes and Decharme, 2012; Decharme et al., 2019). Several studies showed the good performance of ISBA to simulate field sites (Calvet et al., 1998; Boone et al., 2000; Decharme et al., 2011; Joetzjer et al., 2015b; Garrigues et al., 2015, 2018; Morel et al., 2019).

The surface temperature evolves with the heat storage in the soil-vegetation composite and the thermal gradient between surface and the other layers (Boone et al., 2000; Decharme et al., 2011). At depth, the heat transfer is described by the use of the one-dimensional Fourier law. The soil heat capacity is the sum of the water heat capacity and the heat capacity of the soil. The soil thermal conductivity is a function of volumetric water content and porosity. Freezing due to water phase changes in the soil can also be computed by taking into account the effect of the sublimation of frost and the isolation of vegetation at the
surface (Decharme et al., 2016).

ISBA includes an interactive vegetation scheme, activated in this study, that represents 16 broad types of vegetation. The scheme represents plant photosynthesis and respiration, plant growth and decay. The simulated stomatal conductance depends on photosynthesis and controls transpiration. The vegetation canopy is characterized by the leaf area index (LAI), which results



directly from the leaves carbon balance computation. The simulated LAI varies during the year according to photosynthetic activity, respiration and decay. In spring for instance, when photosynthetic activity is high due to high solar radiation and sufficient volumetric water content, LAI increases. LAI affects transpiration and the evaporation of water on the canopy associated to intercepted rain or dew deposition. Photosynthesis and respiration are parametrized according to the semi-empirical model of Goudriaan et al. (1985) and implemented by Jacobs et al. (1996) and Calvet et al. (1998). Plant growth and decay are based

on Calvet and Soussana (2001), Gibelin et al. (2006) and Joetzjer et al. (2015a). A complete description of the carbon cycle in vegetation can be found in Delire et al. (2020).

The ISBA soil hydrology uses the following mixed form of the Richards (1931) equation to describe the water mass transfer within the soil via Darcy's law :

$$\frac{\partial \omega(z)}{\partial t} = \frac{\partial}{\partial z}[-k(z)(\frac{\partial \psi(z)}{\partial z} + 1)] + \frac{S(z)}{\rho_\omega} \tag{1}$$

where $\omega$ ($m^3 \cdot m^{-3}$) is the volumetric water content at each depth $z$ (m), $\psi$ (m) the water pressure head, $k$ ($m.s^{-1}$) the soil hydraulic conductivity. $S$ ($kg.m^{-3} \cdot s^{-1}$) is the soil-water source/sink term especially related to water withdrawn by evapotranspiration in each layers. A complete description of the model equations used to simulate water transport can be found in Boone et al. (2000) and Decharme et al. (2011, 2019). This soil hydrology is solved numerically using a Crank-Nicholson implicit

time scheme where the flux term is linearized via a one-order Taylor series expansion. The resulting linear set of diffusion equations can be cast in a tridiagonal form and solved quickly (cf. Supplement S1 for more details). The soil discretization is adapted to the lysimeter depth and to the measurement depths : 14 soil layers are used, at 0.01, 0.04, 0.1, 0.2, 0.4, 0.6, 1.0, 1.2, 1.4, 1.6, 1.8, 1.95 and 2 m depths. Water for soil evaporation is drawn from the superficial layer. The water used for transpiration is removed throughout the root zone in which the roots are asymptotically distributed according to Jackson et al.

(1996). ISBA uses the soil infiltration at the surface as upper boundary condition, neglecting surface runoff when simulating local field or lysimeter sites. This soil infiltration is the flux of water reaching the soil surface, i.e. the sum of the precipitation not intercepted by the canopy, the dripping from the vegetation and the snow melt. At the bottom of the soil column, a fine layer of 5 cm is used to simulate a seepage face lower boundary condition (LBC) instead of the usual free drainage LBC used in ISBA (cf. Supplement S1.3). Such seepage face LBC is recommended to simulate lysimeters (Séré et al., 2012; Tifafi et al.,

2017). Finally, to compute initial conditions, a spin-up of 1 year is done using the first year of each forcing data in order to ensure an adequate numerical equilibrium for soil water.

## 2.3 Model approaches

In this study, we evaluate four closed-form equations that link volumetric water content, soil matric potential and hydraulic conductivity assuming a residual water content ($\omega_r$) equal to 0. All simulations use the same ISBA configuration and are

denoted as follow :





- **BC66** : this first model approach uses the standard version of ISBA, i.e. the closed-form equations of Brooks and Corey (1966) where the soil water retention function, $\psi(\omega)$, and the hydraulic conductivity function, $k(\psi)$, are given by

$$
\begin{cases}
\psi(\omega) = \psi_{sat} \left( \dfrac{\omega}{\omega_{sat}} \right)^{-b} \\[2mm]
k(\psi) = k_{sat} \left( \dfrac{\psi}{\psi_{sat}} \right)^{-\frac{2b+3}{b}}
\end{cases}
\tag{2}
$$

where $b$ $(-)$ represents the dimensionless shape parameter of the soil water retention function, while $\psi_{sat}$ (m) and $k_{sat}$ (m.s$^{-1}$) are the soil matrix potential and hydraulic conductivity at saturation, respectively.

- **VG80** : this second model approach uses the following closed-form equations from van Genuchten (1980) using Mualem (1976) theory

$$
\begin{cases}
\psi(\omega) = -\dfrac{1}{\alpha} \left[ \left( \dfrac{\omega}{\omega_{sat}} \right)^{-1/m} - 1 \right]^{1/n} \quad \text{with} \quad m = 1 - \dfrac{1}{n} \\[2mm]
k(\psi) = k_{sat} \cdot S^{l}[1 - (1 - S^{1/m})^{m}]^{2} \quad \text{with} \quad S = [1 + |\alpha\psi|^{n}]^{-m}
\end{cases}
\tag{3}
$$

where $\alpha$ (m$^{-1}$) is the inflection point where the slope of the soil water retention function (d$\omega$/d$\psi$) reaches its maximum value, $n$ $(-)$ a dimensionless coefficient that characterizes the shape of the retention curve, and $l$ the Mualem (1976) dimensionless parameter that determines the slope of the hydraulic conductivity function.

- **BCVG** : this third model approach (cf. Fuentes et al. (1992) or Braud et al. (1995)) uses the combination of the $\psi(\omega)$ function from VG80 with the $k(\psi)$ function from BC66 using Burdine (1953) theory as follow

$$
\begin{cases}
\psi(\omega) = -\dfrac{1}{\alpha} \left[ \left( \dfrac{\omega}{\omega_{sat}} \right)^{-1/m_b} - 1 \right]^{1/n_b} \quad \text{with} \quad m_b = 1 - \dfrac{2}{n_b} \\[2mm]
k(\psi) = k_{sat} \cdot S_b^{2\lambda_b + 3} \quad \text{with} \quad \lambda_b = (n_b - 2)^{-1} \quad \text{and} \quad S_b = [1 + |\alpha\psi|^{n_b}]^{-m_b}
\end{cases}
\tag{4}
$$

where $n_b$ $(-)$ is the dimensionless coefficient that characterizes the shape of the retention curve using Burdine (1953) theory.

- **VGc** : this last approach uses the usual $\psi(\omega)$ function from VG80 in Eq. (3) but the corrected form of the $k(\psi)$ function from Iden et al. (2015) using Mualem (1976) theory as follow

$$
k(\psi) = k_{sat} \frac{S^{l}}{\Gamma^{2}} \cdot
\begin{cases}
[1 - (1 - S^{1/m})^{m}]^{2} & \forall \psi \leq \psi_c \\[2mm]
[1 - (1 - S_c^{1/m})^{m} + \frac{S - S_c}{|\alpha\psi_c|}]^{2} & \forall \psi > \psi_c
\end{cases}
\tag{5}
$$

with $S_c = [1 + |\alpha\psi_c|^{n}]^{-m}$ and $\Gamma = 1 - (1 - S_c^{1/m})^{m} + \frac{1 - S_c}{|\alpha\psi_c|}$

where $\psi_c$ (m) is the value of the maximum suction near saturation. It is fixed at -0.01 m, as suggested by Iden et al. (2015), which is equivalent to use a maximum pore-size only in the capillary bundle model.





Additional sensitivity tests were made (see section 5) first to compare the impact of prescribing homogeneous soil profiles, which are classically derived from the use of pedological soil maps, versus heterogeneous soil profiles, which are generally observed in situ. Secondly, we test the performances of using soil parameter values derived from usual PTF (Clapp and Hornberger, 1978; Cosby et al., 1984; Carsel and Parrish, 1988; Wosten and van Genuchten, 1988; Vereecken et al., 1989; Weynants et al., 2009) instead of using soil parameter values observed in situ. Finally, we also assess the impact of using the seepage face LBC compared to the usual free drainage LBC used in ISBA over natural sites and in regional and global scale applications.

## 3  Estimation of parameters

### 3.1  Soil parameters

The rich data sets collected by the lysimeters (hourly resolution with 3220 observations on average by lysimeters and by depth) allow to estimate the soil hydrodynamic parameters, such as in previous studies (Brooks and Corey, 1966; van Genuchten, 1980; Carsel and Parrish, 1988). For instance, Figure 1 plots the observed soil water retention curves, i.e. the volumetric water content, $\omega$, as a function of the logarithm of the absolute value of the soil matrix potential, $\psi$, (dots with different colors for each depth) for lysimeters G2 and O1 (other lysimeters are given in the Supplement Figure S1 and S2). To remove the effect of hysteresis on the observed soil water retention curves, i.e. the difference in soil water retention curve measured under wetting and drying process (Haines, 1930), we have averaged the $\omega$ values for each $\psi$ values. This Figure 1 reveals an important heterogeneity with depth. From these observed soil water retention curves, $\omega_{sat}$, $\psi_{sat}$, $\alpha$, $n$, $n_b$, $b$ can be derived at each depth of each soil column (for example, $b$ is the slope of this relation). To this end, we use an objective least squares function which minimizes the sum of the squares of the deviations. This function allows to maximize the likelihood with a normal distribution (function nls of rstudio). The soil parameters estimates allow to plot soil water retention curves from BC66 and VG80 (red and blue lines, respectively) on Figure 1 showing a better fit for VG80 than BC66.

Next, we use two steps to evaluate this method and the soil parameters estimates. First, we successfully check that (1) the estimated $\omega_{sat}$ is consistent with the 99th percentile of the observed soil volumetric water content at each depth and for each lysimeter ; and that (2) the estimated $n$, $n_b$ and $b$ or $\alpha$ and $\psi_{sat}$ verify the following well-known relationships defined by van Genuchten (1980) :

$$\begin{cases} n \approx 1 + b^{-1} & \text{for Mualem theory} \\ n_b \approx 2 + b^{-1} & \text{for Burdine theory} \\ \alpha \approx |\psi_{sat}|^{-1} \end{cases} \tag{6}$$

Although several authors proposed more complex relationships allowing preservation of the capillary effects (Lenhard et al., 1989; Morel-Seytoux et al., 1996; Leij et al., 2005; Sommer and Stöckle, 2010), a comparison of some of these relationships by Ma et al. (1999) has showed that the van Genuchten (1980) and Lenhard et al. (1989) relationships gave better results. It is interesting to note that the simple relationship $n_b \approx n + 1$ is usually true for our soils (cf. Supplement Table S2). This justifies





our choice not to represent the $\omega(\psi)$ relationship from BCVG (equation 4) on Figure 1 (and thus Supplement Figure S1 and S2) because it is similar to the $\omega(\psi)$ relationship from VG80 (equation 3). Secondly, we compare our estimates to an alternative

method to derive soil parameters from observed soil water retention curves based on the package SoilHyP (Dettmann et al., 2022) which uses the Shuffled Complex Evolution (SCE) optimization. Soil parameters values found with this method are very close to our estimates as shown by low mean relative bias between both estimates (5% for $b$, 1% for $n$, 2% for $\omega_{sat}$, and 4% for $\psi_{sat}$). Note that for $\alpha$ even if the mean relative bias seems to be not negligible (11%), both estimates still remain very close (20.64 $m^{-1}$ on average for SoilHyP and 18.46 $m^{-1}$ for this study).


Estimations of the soil hydraulic conductivity function parameters ($k_{sat}$ and $l$) are more difficult because of the lack of observed hydraulic conductivity profiles in each lysimeter. When the deepest part of each lysimeter is close to saturation, we assume that the soil hydraulic conductivity at saturation, $k_{sat}$, can be considered equal to the observed hourly drainage at 2m depth. $k_{sat}$ is thus derived empirically from the 99.9 percentile of the hourly drainage distributions. No assumption is done on

an hypothetical $k_{sat}$ profile with depth, i.e. $k_{sat}$ is taken homogeneous with depth in each lysimeter. For VG80, a particular attention is also given to the very low $n$ values for which (Eq.3) is numerically unstable. In this study, we found that our simulations are numerically stable when $n \geq 1.1$, so the limit of $n$ is fixed at $n = 1.1$. As already said, BCVG (Eq.4) and VGc (Eq.5) allow to cancel this limitation. The $l$ parameter in Eq. (3) for VG80 and Eq. (5) for VGc is estimated with a simple calibration via ISBA sensitivity tests with $l$ ranging from -5 to 5. Better scores are obtained for $l$ equal to 0.5, a classical value, for the

OPE experimental station. For the GISFI experimental station, better scores are obtained for $l$ equal to 0.5 (G4), -2 (G1,G2) and -5 (G3) which remains consistent with the literature (Wosten and van Genuchten, 1988; Wösten et al., 2001; Schaap et al., 2001). We choose $l$ to be constant over the entire soil profile for each lysimeter.

Figure 2 gives an example for lysimeters O1 and G2 of the relative soil hydraulic conductivity, $k/k_{sat}$, as a function of the

soil water content actual saturation, $\omega/\omega_{sat}$, derived from the four closed form equations described previously (other lysimeters are given in Supplement Figure S3). The VG80 hydraulic conductivity exhibits an abrupt drop at the transition from saturated conditions to unsaturated conditions, contrary to BC66. This behaviour is well known for fine-textured soil as in our lysimeters (Fuentes et al., 1992; Vogel et al., 2000; Ippisch et al., 2006; Iden et al., 2015). This explains why the VG80 hydraulic conductivity function with small $n$ values is strongly unstable near saturation. The comparison with the observed hourly drainage

water at 2 m depth (reduced to $k_{sat}$) versus the actual saturation at 1.5 m depth seems to confirm that VG80 certainly underestimates the actual hydraulic conductivity near saturation for fine-textured soil as underlined by previous studies (Vogel et al., 2000; Ippisch et al., 2006; Iden et al., 2015).

All parameter estimates are presented in Figure 3 and in the Supplement Table (S2). Parameters vary greatly with depth

(especially $b$, $n$, $\psi_{sat}$ and $\alpha$) and, to a lesser extent, between lysimeters. $n_b$ is not shown on Figure 3 because its varies like $n$. Although these heterogeneities can be observed on field sites, they could also be increased by factors like compaction and structuration generally present on lysimeters (Weihermüller et al., 2007; Séré et al., 2012). The largest differences are found for



$b$ (and thus $n$). For all lysimeters except G1 and G4, $b$ increases with depth (and conversely $n$ decreases). For the OPE lysimeters, $b$ starts from 8 at 0.2 m depth, doubles first at 0.5 m and then at 1 m depth, and reaches 50 at 1.5 m. These variations are

less pronounced for the GISFI's lysimeters, with only a variation with depth by a factor of two for G2. Absolute value of $\psi_{sat}$ decreases with depth (and conversely $\alpha$ increases) especially for the OPE lysimeters. Lysimeters were filled with preserved soil columns at the OPE, and manually at the GISFI. Repacked soil columns are generally recognised more homogenous than undisturbed monoliths (Weihermüller et al., 2007; Carrick et al., 2017). The fact that the GISFI lysimeters are filled in a manual way can thus explain weaker soil heterogeneities with depth, although they have been filled to preserve the bulk density of the

soil.

Comparing our parameter estimates to PTFs usually used in LSMs (Clapp and Hornberger, 1978; Cosby et al., 1984; Carsel and Parrish, 1988; Wosten and van Genuchten, 1988; Vereecken et al., 1989; Weynants et al., 2009), we find that $b$ (and thus $n$) from in situ estimates are quite different from those determined by the PTFs (boxplots). While $b$ ranges from 5 to 12 with

the usual PTFs, our estimates range from 8 to 50. $\psi_{sat}$ (and thus $\alpha$) estimates are slightly different from those determined by usual PTFs. For other parameters, our estimates and the usual PTFs give similar values.

### 3.2 Vegetation parameters

For the lysimeters covered by vegetation (O1, O2, O3, G3 and G4), two additional soil parameters must be determined. The

field capacity $\omega_{fc}$ and the wilting point $\omega_{wilt}$ are computed via matrix pressure at -0.33 bar and -15 bar, respectively. The root depths have also been determined. Although rooting depth was not measured at the sites, it is possible to derive it from the observations of the volumetric water content profile: if the volumetric water content presents a slow decrease in summer at a given depth, it is considered that the roots have not yet reached this depth. The root depth is thus fixed at 2 m for lysimeters G3 and O2, and varies for lysimeters G4, O1 and O3. From 2009 to 2013, root depth in lysimeter G4 reached 1.5 m; but after June

2013, seeding and harvesting were carried out every year limiting root development that never reached below 0.4 m depth. In lysimeters O1 and O3, the root depth reached 0.8 m from 2014 to 2018, and 2 m thereafter. Standard ISBA values of maximum photosynthesis, leaf nitrogen content and specific leaf area were used for the lysimeters covered by grass, but specific values were derived form the TRY database (Kattge et al., 2020) for alfalfa as this crop is not a standard vegetation type in ISBA.

As there are no measurements of LAI for these lysimeters, the simulation of the LAI can only be compared to the literature. As expected, the simulated LAI is minimal between December and April and maximal in August, and they are variabilities between years (Supplement Figure S4). For soils with grass (G4, O1, O2, and O3), the maximum LAI varies between 3 and 5, and the mean annual LAI varies between 1.3 and 2. These values are similar to those found in other studies (Calvet, 2000; Darvishzadeh et al., 2008). LAI for alfalfa cover (G3) is larger ($LAImax = 7.5$, $LAImean = 4.8$) which is expected for such

well-developed vegetation (Wolf et al., 1976; Wafa et al., 2018).




# 4 Results

Here, we present the main results for each model approach described in section 2.3 in terms of simulated soil water and drainage water dynamics, water budget and intense drainage water events. We used four skill scores to assess the ability of each model approach to reproduce. The overall bias between simulatuion and observation, the centred root mean square error ($CRMSE$) computed by subtracting the simulated and observed annual means from their respective time series before computing a standard root mean square error, the coefficient of determination ($R^2$) or the Pearson correlation ($R$), and the Nash and Sutcliffe (1970) efficiency criterion ($NASH$) that determines the relative magnitude of the residual variance compared to the measured data variance. These score are summarized in Appendix A. Note that the simulated soil temperatures have also been studied and analysed. All model approaches have shown good skill scores ($R^2 > 0.9$) underlying the ability of the ISBA LSM in reproducing observed soil temperatures. Because there are no significant differences between the four model approaches compared to the observations, these results are not presented in this study.

## 4.1 Soil water dynamics

The dataset allows assessing the evolution of the total soil water mass derived from the mass of the lysimeter, of the soil volumetric water content at several depths, and of the drainage water flux at the bottom of the lysimeters. In the following analysis, periods when the meteorological forcing is reconstructed or when data is of poor quality (Supplemen Table S1) are not taken into account in the scores computation. These periods are short, except for the drainage water of lysimeter G3 (23 % of the duration).

### 4.1.1 Water mass time variations

As there is no observations of the weight of the dry soil in each lysimeter that can be serve as reference, we evaluate the ability of ISBA to simulate the temporal variation of water mass around the mean and not around the absolute mass of each lysimeter. These variations are assumed to be equal to the time variations of the total mass of the lysimeter, neglecting the seasonal variations of the vegetation mass. Time variations of the water masses are presented in Figure 4 for the seven lysimeters at 1-hour time step. All model approaches (BC66, VG80, BCVG and VGc) are represented.

With soil parameters determined in situ, the evolution of the total soil water mass is well reproduced by ISBA whatever the model approach (Figure 4). Skill scores (Figure 5) are better for OPE experimental station lysimeters with small $CRMSE$ (below 56 kg, except for O1 with VG80). The comparison of the four model approaches shows that BC66, BCVG, and VGC exhibit better scores with mean $CRMSE$ of 32 kg, 32 kg and 34 kg respectively compared to 43 kg for VG80. VG80 exhibits also the lower $R^2$ (0.68) compared to other experiments (0.75, 0.75 and 0.72 for BC66, BCVG and VGC, respectively).





### 4.1.2 Water Content


Time evolutions of the water content at 0.5 m are shown in Figure 6 while the scores are presented for the 2 depths for which there are the most usable observations, i.e 0.5 m and 1 m (Figure 5). For lysimeters with vegetation (G3, G4, O1, O2, and O3), the roots draw water in the summer period which reduces the volumetric water content, causing a more pronounced contrast in volumetric water content between winter and summer than for bare soil lysimeters. This behavior is well represented by ISBA:

biases are negligible ($<6.\ 10^{-2}\ \mathrm{m^3.m^{-3}}$) and dynamics are correct ($R^2 >0.5$, not shown). Still minor differences between BC66, BCVG and VGc appear. VG80 obtains weaker statistical scores in 65 % of the cases, because soil water saturation is reached too rapidly. VGc is clearly a solution to simulate water content when the parameter $n$ is closed to 1, by correcting the default of VG80 for these soils.

It should be noticed that agreement between observed and simulated soil volumetric water contents is weaker at the shallow depth available only at OPE (0.2 m) than at the other depths with $^2 < 0.6$ (not shown). This can be explained by the different processes which can modify the structure of the soil at the surface: the intensity of precipitation can increase the soil surface sealing (Liu et al., 2011; Assouline, 2004), and the soil heterogeneity can increase in response to plant or biological activities (Brown et al., 2000; Beven and Germann, 1982). Such processes are not represented in ISBA.

### 4.1.3 Drainage water


Drainage at 2 m depth is measured at an hourly frequency. However, to compensate for the measurement limits associated with a $0.1\ \mathrm{m.h^{-1}}$ threshold, the data is aggregated daily. The annual volume drained varies significantly between lysimeters of the same sites (Supplement Table S2), although it is assumed they are exposed to the same atmospheric conditions. On the GISFI experimental station, the mean annual drained water is maximum on bare soil lysimeters G1 and G2 ($>300\ \mathrm{mm.year^{-1}}$). The

mean annual drained water is two to three times lower for the lysimeters covered by vegetation. At the OPE experimental station, all the lysimeters have a vegetation cover, and the mean annual drained water shows a variation of only 16 % from 300 to 363 $\mathrm{mm.year^{-1}}$. Such amounts are comparable to the volume drained on the bare soil of the GISFI experimental station, which is mainly due to higher mean annual precipitation at OPE.

Daily mean annual cycles of drainage water are shown in Figure 7. At the GISFI experimental station, drainage water occurs almost all year long for bare soil lysimeters G1 and G2. The well developed vegetation cover in G3 causes a decrease in both drainage water intensity and drainage water duration. At the OPE experimental station, the drainage water occurs mainly from October to June (if the year 2016 is excluded, as a large rainfall event occurred in May-June 2016), with similar cycles for the 3 lysimeters. The annual cycle is well simulated, with more discrepancies for the VG80 model approach, which tends to

overestimate the drainage water during some recession periods (as for example during spring for G1, G4 and O3).





Daily drainage water is shown in Figure 8 and the scores are given in Figure 5. Even if the annual volume drained is higher at the OPE lysimeters, the maximum drainage water intensities over the observed period are similar at both sites: they vary between 27.4 and 34.0 $\mathrm{mm.day}^{-1}$ at OPE experimental station and between 24.0 and 33.0 $\mathrm{mm.day}^{-1}$ at the GISFI site.

The four model approaches reproduce the daily drainage water with relatively low biases ($<0.7$ $\mathrm{mm.day}^{-1}$), worst biases being obtained by VG80 especially on the GISFI lysimeters and confirming results shown in Figure 7. Dynamics are also well reproduced as shown by the $NASH$ scores. These scores are similar for BC66, BCVG and VGc with an average $NASH$ of 0.42, 0.42 and 0.43, respectively, but slightly lower score,0.35, for VG80. This worst performance of VG80 to reproduce daily drainage water is especially true for the GISFI lysimeters.

## 4.2 Water Budget

Lysimeters give access to an estimate of the actual evapotranspiration (Schrader et al., 2013; Gebler et al., 2015). Neglecting lateral runoff (not present in these lysimeters), the following simple water balance equation allows to estimate annual evapotranspiration ($E$) from annual precipitation ($P$), annual drainage ($Q_{drain}$) and a water mass variation negligible ($\Delta W$) for each lysimeter:

$$E = P - Q_{drain} - \Delta W \tag{7}$$

Figure 9 presents the water budget observed and simulated over the entire period for all lysimeters. The total evapotranspiration and the drainage water ratios to the total precipitation are expressed in percentage. At the GISFI experimental station, between 50 and 80 % of the annual rainfall is evapotranspired, with maximal uptake on lysimeter G3 with the densest vegetation cover. At the OPE experimental station, evapotranspiration corresponds to nearly 50 % of the rainfall.

The annual budgets simulated by ISBA are rather close to the observations, but the drainage water is generally overestimated and thus the evapotranspiration underestimated on all lysimeters. The absolute difference averaged over all the lysimeters is lower for BC66 (8.1 percentage point) compared to the other model approaches (10.8 percentage point for BCVG, 11.7 for VGc and 12.9 for VG80). This seems to underline that using closed-form equations from van Genuchten (1980) could favour drainage water at the expense of evapotranspiration compared to Brooks and Corey (1966), at least over our lysimeters and

with the ISBA LSM.

### 4.3 Intense Drainage events

In the previous sections (4.1.3, 4.2), drainage water is analyzed on complete chronicles, where strong daily drainage water events were detected. In order to check the ability of the four model approaches to reproduce strong soil water dynamic, a focus is made on intense drainage water events. All daily drainage water fluxes larger or equal to the 99th percentile of the

daily drainage water distribution over the entire period are selected for each lysimeter. These Q99 values are higher for OPE experimental station lysimeters ($>13$ $\mathrm{mm.day}^{-1}$) than for GISFI experimental station lysimeters (5.4 to 9 $\mathrm{mm.day}^{-1}$). A total of 110 events on the set of the 7 lysimeters is selected. 75 % of these intense drainage water events appear from October to March, i.e. during the wet period when the soil is near saturation. The remainder occurs in May and June associated with



intense precipitation events as generally observed in this region.


Figure 10 presents winter intense drainage water events in February 2016 for two contrasted lysimeters (O1 with vegetation and G2 with bare soil). February 2016 corresponds to a period of approximately one month (31 days for O1 and 27 days for G2), with some daily precipitations above 20 mm.day$^{-1}$, and an initially wet soil. Figure 11 presents late spring events in June 2016 that led to a flood event of the Seine river (Philip et al., 2018). This event is characterised by large accumulated precipitation (166 mm at OPE and 210 mm at GISFI in 10 days) and intense daily precipitation with a maximum on May, 30th that reached above 40 mm.day$^{-1}$ and 70 mm.day$^{-1}$ at the OPE and GISFI experimental stations, respectively. Figure 10 and 11 show observed daily precipitation, hourly observed and simulated soil profile saturation, and daily observed and simulated drainage water. VG80 tends to simulate too wet conditions over the entire soil profile compared to the observations. On these lysimeters, BC66, BCVG and VGc reproduce well soil moisture profiles. The drainage water simulated with VG80 further from observations than with other model approaches. In one case (G2 lysimeter) this VG80 simulated drainage water is occurring too early whatever the season, while in the other case (O1 lysimeter) its dynamic is too smooth during winter. Conversely, the dynamic of these events are well reproduced in phase and maximum intensity by BC66, BCVG and VGc, as highlighted by the good $NASH$ scores. When the same comparison is made on the 110 selected intense drainage water events, the scores are significantly better for BCVG and VGc. They exhibit the lowest bias (1.11 and 0.8 mm.day$^{-1}$) compared to the other model approaches (1.3 mm.day$^{-1}$ for BC66 and 1.26 mm.day$^{-1}$ for VG80), as well as highest $NASH$ criteria (0.70 and 0.80 for BCVG and VGc compared to 0.70 and 0.56 for BC66 and VG80).

### 4.4 Synthesis

To summarize the results, Taylor diagrams (Taylor, 2001) are used to quantify the degree of correspondence between the modeled and observed behavior in terms of three statistics: the Pearson correlation coefficient, the centred root-mean-square error, and the normalized standard deviation (Figure 12). These scores are computed using all the seven lysimeter time series, with a single result for BC66, VG80, BCVG and VGc.

For water mass and volumetric water content (Figure 12a and b), scores are calculated at an hourly time-step, while a daily time step is used for the drainage water over both the full period and the 110 drainage water intense events (Figure 12 c and d). BC66, BCVG and VGc obtain good results, especially to predict water mass, volumetric water content and drainage water during intense events. Consistently with previous results, VG80 obtains significantly lower scores whatever the observable. VGc exhibits the larger score in term of intense drainage events and the same scores than BC66 and BCVG for other variables. These results highlight the VGc model of Iden et al. (2015) as a very interesting alternative to the VG80 model for hydrological applications with a LSM while maintaining an approach integrally based on closed-form equations from van Genuchten (1980).





# 5 Sensitivity model approaches

## 5.1 Homogeneous soil profile

As LSMs are used on regional to global scales, their soil parameters are usually derived from soil maps that generally consider an homogeneous soil profile. To evaluate the influence of the variation with depth of soil hydrodynamic parameters on our simulations we performed a sensitivity test with uniform soil profile using the BC66 model (BC66$_{HOM}$) and the VGc model (VGc$_{HOM}$). This uniform profile is fed with the vertical mean value of each parameter for each lysimeter (cross on Figure 3). Using an homogeneous profile in BC66$_{HOM}$ and VGC$_{HOM}$ significantly degrades the scores in terms of water content (Figure 12) but have a limited impact on the simulated water budget compared to BC66 and VGc (Figure 9). It has a stronger impact on intense drainage events than on the whole time series (Figure 12 and Table 2). Figure 13 shows very clearly that BC66$_{HOM}$ fails to capture the observed soil moisture profile (drier surface conditions and wetter in depth), conversely to BC66 (Figures 10 and 11). VGc$_{HOM}$ exhibits the same behavior (Supplement Figure S5). The drainage water dynamics is less impacted during winter than during spring as shown by the BC66$_{HOM}$ $NASH$ scores compared to BC66 (and accordingly for VGC$_{HOM}$ and VGc). Indeed, lysimeters soils simulated with an homogeneous profile appear wetter than observations and reference simulations, especially in spring. This wetter state induces logically a too intense reactivity of the drainage water during this period, reducing the simulated skill scores compared to reference simulations. This fact underlines that using an homogeneous soil profile fails sometime and for some conditions to correctly simulate the drainage dynamic in the studied lysimeters.

In order to determine which hydro-dynamical parameter via its vertical profile has the largest impact on the simulations, we performed additional sensitivity tests with homogeneous soil profiles for all parameters except for one that keeps its estimated heterogeneous profile. These tests are performed with all parameters homogeneous except one, either $\omega_{sat}$, $b$, or $\psi_{sat}$ for BC66 and $\omega_{sat}$, $n$, or $\alpha$ for VGc.

Using the seven lysimeters complete drainage water time series and all the selected intense drainage water events as in section 4.4, Table 2 shows that $b$ and $n$ are the most important parameters as accounting for their heterogeneous profile improve more the $NASH$ score compare to the other parameters. Their $NASH$ are in addition very close to the BC66 and VGc references. These tests demonstrate the importance of $b$ (and therefore $n$) to accurately simulate the drainage water dynamic and intense drainage water events. This finding is in agreement with previous studies (Ritter et al., 2003) that demonstrated a strong sensitivity of the simulated drainage water to $n$ (and therefore $b$) and a lower sensitivity to $k_{sat}$.

## 5.2 Usual pedotransfer functions

LSMs commonly use PTF to derive soil hydrodynamic parameters from soil textural information. As shown in Figure 3, the soil parameters estimated from our measurements can be very different from those derived from six usual PTF (Clapp and Hornberger, 1978; Cosby et al., 1984; Carsel and Parrish, 1988; Wosten and van Genuchten, 1988; Vereecken et al., 1989; Weynants et al., 2009). This is especially true for the $b$ and $n$ parameters. To investigate the impacts of such differences, sensi-





tivity tests noted $BC66_{PTF}$ and $VGc_{PTF}$ are performed in which the soil hydrodynamic parameters in each soil horizon are derived from the mean of these PTFs. Impacts on the simulated water budget are clear but not obvious to comment (Figure 9), even if $BC66_{PTF}$ and $VGc_{PTF}$ tend to increase drainage water at the expense of evapotranspiration compared to BC66 and VGc over the GISFI lysimeters, while the opposite behavior (lower drainage water and larger evapotranspiration) is found for the OPE lysimeters. All skill scores are drastically degraded in terms of water mass variations, volumetric water content at 0.5m depth, daily drainage water and intense drainage events (Figure 12 and Table 2). These weaknesses are highlighted over the selected February and June 2016 events (Figure 13 and Supplement Figure S5). The soil water profile simulated by $BC66_{PTF}$ and $VGc_{PTF}$ is strongly underestimated compared to observations, especially for the G2 lysimeter. This weakness induces a significant delay of the simulated drainage water compared to observations and to other simulations.

Once again, the $b$ and $n$ parameters seem to be the keys to this weakness. Indeed, we performed a set of tests using the $BC66_{PTF}$ and $VGc_{PTF}$ configuration except for one parameter, either $b$ (or $n$), $\omega_{sat}$, $\psi_{sat}$ (or $\alpha$), or $k_{sat}$, that keeps its estimated in situ value. The use of the in situ estimated values of $b$ and $n$ tends to balance the water budget toward its partitioning of reference (Figure 9), especially with VGc. It drastically improves all skill scores compared to $BC66_{PTF}$ and $VGc_{PTF}$ (Figure 12). This is also the case for the simulation of the soil water profile and especially the soil drainage water dynamic (Table 2, Figure 13, and Supplement Figure S5).

## 5.3 Lower boundary conditions

LSMs commonly use free drainage LBC on field sites or in regional and global scale application. We can wonder if such LBC is able to impact our results compared to the seepage face LBC classically used to simulate lysimeters. Four simulations were performed using the same four model approaches as previously but with a free drainage LBC instead of a seepage face LBC.

Figure 14 shows the mean daily annual cycles of the soil moisture profile and the drainage water observed and simulated by BC66 and VGc with the two LBC approaches ($BC66_{FREE}$ and $VGc_{FREE}$ compared to $BC66_{SEEPAGE}$ and $VGc_{SEEPAGE}$) for O1 and G2 lysimeters. Soil moisture profiles are sensibly the sames even if, logically, the deep layers are slightly drier because the deepest layer is always saturated with a seepage face LBC. This fact is confirmed for all model approaches (BC66, VG80, BCVG and VGc) which exhibit the same skill score in term of water mass variations and volumetric water content at 0.5m whatever the LBC (comparing Figure 12 and Supplement Figure S6). The simulated total water budgets are also relatively unchanged (Supplement Figure S7).

In terms of daily drainage, the simulated response is not very different whatever the LBC (Figure 14). To use a seepage face LBC seems effectively more adequate to simulate drainage water in our lysimeters during recession periods, especially with BC66 (that explain the larger $NASH$ for $BC66_{SEEPAGE}$ compared to $BC66_{FREE}$ for the G2 lysimeter). Some peaks of drainage water are also more pronounced with such LBC. Comparing Taylor diagrams (Figure 12 and Supplement Figure




S6) underlines that all model approaches (BC66, VG80, BCVG and VGc) reproduce daily drainage water and intense drainage events with the same accuracy whatever the LBC.

## 6 Discussion and Conclusion

This study uses time series (up to 7 years) from several lysimeters to evaluate the dynamics of water transfer in the unsatu-
rated zone simulated with the land surface model ISBA. These observations allow deriving the heterogeneous profile of soil parameters. Although the original version of ISBA performed well, a set of four water closure relationships, which estimate the evolution of soil hydrodynamic properties with soil moisture are tested. The comparison of these four relationships shows that, when soil parameters and meteorologic forcing are known, ISBA reproduces the evolution of soil hydrology and vegetation processes reasonably well. The choice to use a seepage face LBC in ISBA (as commonly done to simulate lysimeters) instead
of its usual free drainage LBC has a very little impact on the results.

The simulation using the VG80 water closure relationships exhibits more difficulty reproducing the soil water profile and the drainage water dynamic, in particular during intense drainage water events, than the original ISBA version using the BC66 equations. It is partly linked to the limitation of the VG80 hydraulic conductivity function for n close to 1. The BCVG model
approach that combines the soil matrix potential function from VG80 with the hydraulic conductivity function from BC66, and the VGc model approach of Iden et al. (2015) that includes a maximum pore radius on the VG80 hydraulic conductivity curve solve these problems and even seem to be able to improve the simulation of the soil drainage water dynamic compared to BC66. However, the $l$ parameter in the VG80 hydraulic conductivity (equation 3), and thus VGc (equation 5), is difficult to set even with direct observation. In our study, it is fixed at the classical 0.5 value for some lysimeters, but can vary drastically in
others consistently with the literature (Wosten and van Genuchten, 1988; Wösten et al., 2001; Schaap et al., 2001), underlying the difficulty to estimate this parameter for regional to global scale applications.

The observations show that the soil hydrodynamic parameters in each lysimeter are strongly heterogeneous with depth, while LSMs generally use homogeneous profiles. Using additional sensitivity tests with such homogeneous profiles, we found that
even if the simulated soil water and drainage water dynamics remain acceptable compared to the observations, all the skill scores are worsened (especially for the soil water profile) compared to the model approaches with an heterogeneous profile. This finding support the need to account for vertical heterogeneity of soil hydrodynamic parameters (King et al., 1999; Mirus, 2015; Hengl et al., 2017; Vogel, 2019; Fatichi et al., 2020; Bauser et al., 2020; Gebler et al., 2017) to improve the simulation of soil water and drainage water dynamics (Stieglitz et al., 1997; Mohanty and Zhu, 2007; Decharme et al., 2011; Vereecken
et al., 2019). This is a challenge to simulate groundwater recharge on regional and global scales.

We also found that parameters $b$ and $n$, which characterizes the shape of the soil water retention function, derived from the observations significantly differ from those derived from PTF commonly used in LSMs (Clapp and Hornberger, 1978; Cosby





et al., 1984; Carsel and Parrish, 1988; Wosten and van Genuchten, 1988; Vereecken et al., 1989; Weynants et al., 2009). Sen-
sitivity tests show that the values of $b$ and $n$ derived from usual PTF are not suitable to simulate the drainage water dynamic,
at least over the 7 lysimeters used in this study. In addition, these parameters exhibit the largest heterogeneity with soil depth.
Neglecting this behavior contributes to degrade the simulated drainage water dynamic. Note that this heterogeneous behavior
of $b$ and $n$ is still under consideration in the literature. As in our study, some authors observed a decrease of $n$ with soil depth
(Ritter et al., 2003; Jhorar et al., 2004; Schwärzel et al., 2006), while some others showed an increase (Groh et al., 2018) or
even no change (Schneider et al., 2021). In any case, $b$ and $n$ could be key parameters to correctly simulate drainage water
dynamic and groundwater recharge with LSMs. We recognise that this last assumption has to be confirmed over many other
experimental field sites. Indeed, this study is based on two experimental sites with similar climatic conditions, with low inten-
sity precipitation events compared to other regions. It would be interesting to conduct additional studies in other contrasting
climates.


Finally, this study increases the confidence that LSMs are powerful tools to simulate the recharge of groundwater, in different
environmental conditions, with many soils and vegetation covers, and therefore can be used for many applications in hydrology
at both the regional and the global scales. The sensibility of our LSM to the soil heterogeneity or the value of some hydro-
dynamical parameters underline however that this remains a challenge even with the advent of global databases describing the
vertical profile of the soil properties at depths greater than 1m (Poggio et al., 2021).

## Appendix A:  Statistical scores

In this study, we used the following skill scores considering the simulated, $S_i$, and the observed, $O_i$, data defined at N discrete
points in time :

– The overall bias :

$$Bias = \overline{S} - \overline{O} \quad \text{with} \quad \overline{S} = \frac{1}{N} \sum_{i=1}^{N} S_i \quad \text{and} \quad \overline{O} = \frac{1}{N} \sum_{i=1}^{N} O_i \tag{A1}$$

– The centred root mean square error :

$$CRMSE = \sqrt{\frac{1}{N} \sum_{i=1}^{N} [(S_i - \overline{S}) - (O_i - \overline{O})]^2} \tag{A2}$$

– The Pearson correlation coefficient (or the coefficient of determination) :

$$R = \frac{\frac{1}{N} \sum_{i=1}^{N} [(S_i - \overline{S}) - (O_i - \overline{O})]}{\sigma_s \sigma_o} \tag{A3}$$

where $\sigma_s$ and $\sigma_o$ are the standard deviations of $S_i$ and $O_i$, respectively. The coefficient of determination, $R^2$, is the
square of the Pearson correlation.



– The Nash and Sutcliffe (1970) efficiency criterion :

$$NASH = 1 - \frac{\sum_{i=1}^{N}(S_i - O_i)^2}{\sum_{i=1}^{N}(O_i - \overline{O})^2} \tag{A4}$$

*Author contributions.* The article was written by AS with contributions from all co-authors. AS, BD, and FH developed the study design. NE, POR, and PFC collected the lysimeter data and assisted in their use. AS and BD developed new parameterizations. AS analyzed the data and ran the models with general support from BD and FH. Specific supports was provided by CD for parameterizing vegetation and by PLM to compute weather forcing.

*Competing interests.* The authors declare that they have no conflict of interest.

*Acknowledgements.* The authors are very grateful to Corinne Leyval from LIEC Nancy and Geoffroy Séré from LSE for all the useful discussions and their willingness to provide access to the data sets from GISFI. The authors thank the GISFI for providing field site data at the experimental station in Homécourt and in particular the technical staff, Cyrielle Boone. The OPE dataset was provided by ANDRA with the help of Catherine Galy from ANDRA. Thanks to the IR OZCAR infrastructure for enabling the connection between research from different topics at the various sites. The primary author is funded by the "Centre National de Recherches Météorologiques" of Météo-France,
the "Agence de l'Eau Seine Normandie" and the "Laboratoire de Géologie de l'Ecole Normale Supérieure" of Paris.



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



**Table 1.** Description of each lysimeter: filling method, soil type, vegetation cover, number of texture observations, and textures (in % of clay, sand and silt) at different depths.

| Experimental station | GISFI | | | | OPE | | |
|---|---|---|---|---|---|---|---|
| **Lysimeters** | **G1** | **G2** | **G3** | **G4** | **O1** | **O2** | **O3** |
| Filling method | Fill | Fill | Fill | Monolith | Monolith | Monolith | Monolith |
| Soil | Technosol | Technosol | Technosol | Cambisol | HyCa* | Cambisol | HyCa* |
| USDA soil type | Sandy Clay Loam | Sandy Clay Loam | Sandy Clay Loam | Clay Loam | Clay | Clay Loam | Silty Clay Loam |
| Soil cover | bare soil | bare soil | Alfalfa | Grass | Grass | Grass | Grass |
| Layers | 1 | 1 | 1 | 4 | 6 | 1 | 6 |
| Bulk density (kg.m$^{-3}$) | 1300 | 1300 | 1300 | 1300 | 1700 | 1700 | 1700 |
| **Soil texture profile** (%) | Sand, Clay, Silt | Sand, Clay, Silt | Sand, Clay, Silt | Sand, Clay, Silt | Sand, Clay, Silt | Sand, Clay, Silt | Sand, Clay, Silt |
| Homogeneous | 61.6, 14.3, 24.1 | 61.6, 14.3, 24.1 | 62.4, 15.2, 22.4 | 32.0, 25.0, 43.0 | 3.0, 36.0, 61.0 | 31.0, 41.0, 28.0 | 18.0, 47.0, 36.0 |
| 0.2m | "" | "" | "" | 20.0, 15.0, 75.0 | 11.0, 4.0, 85.0 | 50.4, 18.0, 31.6 | 24.0, 28.0, 48.0 |
| 0.5m | "" | "" | "" | 17.0, 26.0, 57.0 | 0.0, 67.0, 37.0 | 42.0, 27.0, 31.0 | 16.0, 53.0, 31.0 |
| 1m | "" | "" | "" | 34.0, 33.0, 33.0 | 0.0, 19.0, 81.0 | 22.0, 6.0, 72.0 | 16.0, 53.0, 31.0 |
| 1.5m | "" | "" | "" | 56.0, 24.0, 20.0 | 0.0, 19.0, 81.0 | 22.0, 6.0, 72.0 | 16.0, 53.0, 31.0 |

*Hypereutric Cambisol





**Table 2.** $NASH$ scores for the simulated drainage water ($Q_{drain}$) over the 7 lysimeters and the entire period, and during intense drainage water events ($Q_{int}$). Model approaches are shown with soil hydrodynamic parameters set with an homogeneous vertical profile (BC66$_{HOM}$ and VGc$_{HOM}$) or computed using usual PTF (BC66$_{PTF}$ and VGc$_{PTF}$), or derived from observation (BC66, VGc). The $NASH$ of the additional model approaches with one parameter ($n$, $\psi_{sat}$, $\omega_{sat}$, or $k_{sat}$) that keeps the reference values are also given.

| $NASH$ | BC66$_{HOM}$ | | BC66$_{PTF}$ | | BC66 (ref.) | |
|---|---|---|---|---|---|---|
| | $Q_{drain}$ | $Q_{int}$ | $Q_{drain}$ | $Q_{int}$ | $Q_{drain}$ | $Q_{int}$ |
| | 0.34 | 0.60 | -0.08 | -0.25 | 0.42 | 0.70 |
| $b$ | 0.38 | 0.70 | 0.29 | 0.78 | | |
| $\psi_{sat}$ | 0.35 | 0.67 | -0.10 | -0.60 | | |
| $\omega_{sat}$ | 0.34 | 0.62 | -0.10 | -0.20 | | |
| $k_{sat}$ | NA | NA | -0.10 | -0.45 | | |
| | VGc$_{HOM}$ | | VGc$_{PTF}$ | | VGc (ref.) | |
| | 0.41 | 0.65 | 0.08 | -0.12 | 0.45 | 0.80 |
| $n$ | 0.43 | 0.75 | 0.39 | 0.85 | | |
| $\alpha$ | 0.41 | 0.72 | -0.03 | -0.46 | | |
| $\omega_{sat}$ | 0.43 | 0.72 | 0.08 | -0.20 | | |
| $k_{sat}$ | NA | NA | 0.10 | -0.14 | | |

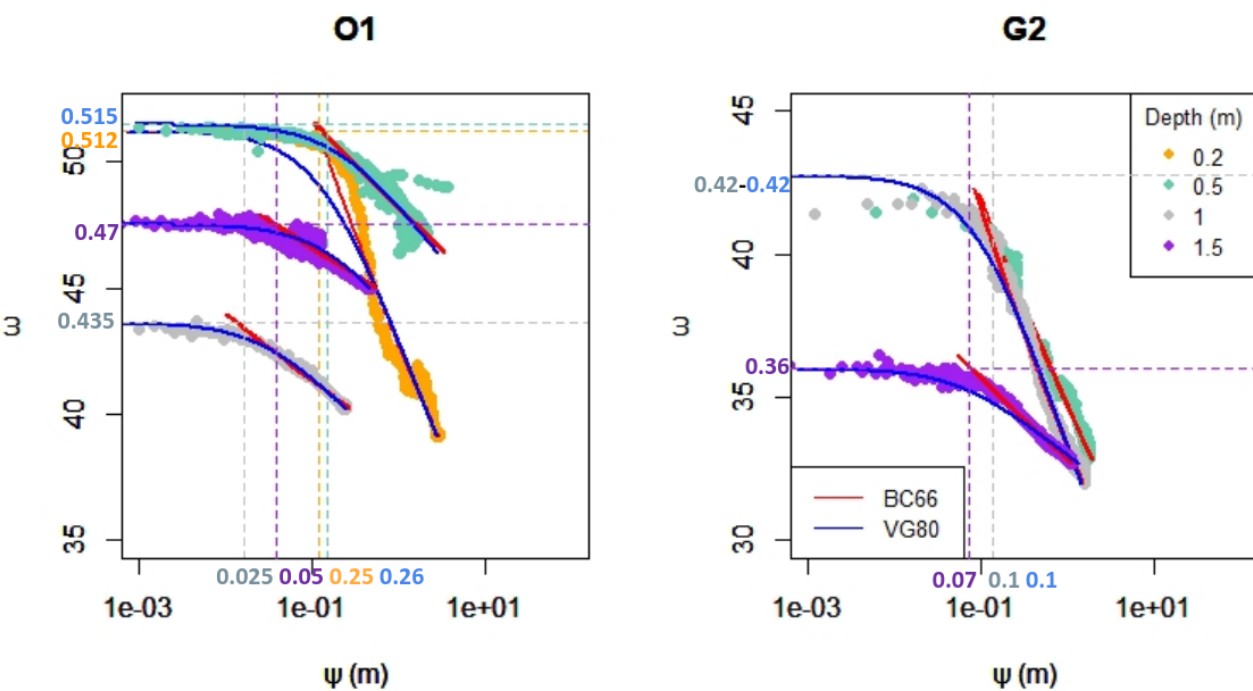

**Figure 1.** Soil water retention curves after removing the effect of hysteresis : volumetric water content ($\omega$) and logarithm of the absolute value of the soil matric potential ($\psi$) for lysimeters O1 and G2. Observations at 0.2, 0.5, 1.0 and 1.5 m depth are represented by orange, aquamarine, grey and purple diamonds respectively, and estimations by red and blue dots for BC66 and VG80, respectively. The dashed lines are the estimated values via observations for the water content at saturation and matric potential.





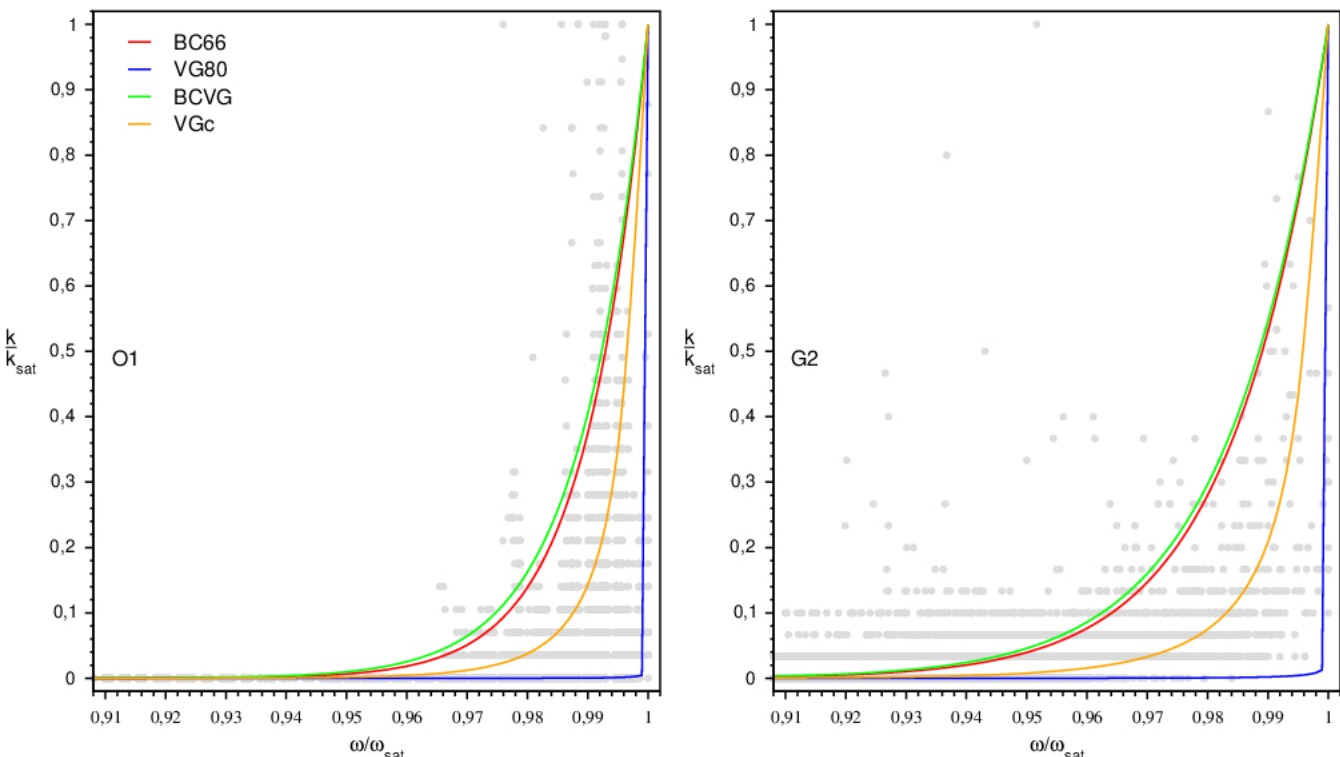

**Figure 2.** Near saturation estimates of the relative soil hydraulic conductivity, $k/k_{sat}$, as a function of the soil water content actual saturation, $\omega/\omega_{sat}$, for lysimeters O1 and G2 at 1.5 m depth. BC66 (red), VG80 (blue), BCVG (green), VGc (orange) are computed using parameter estimates (Supplement Table S2) into Eq.2, Eq.3, Eq.4 and Eq.5, respectively. The dots represent the observed hourly drainage water at 2 m depth (reduced to $k_{sat}$) versus the actual saturation at 1.5 m depth.




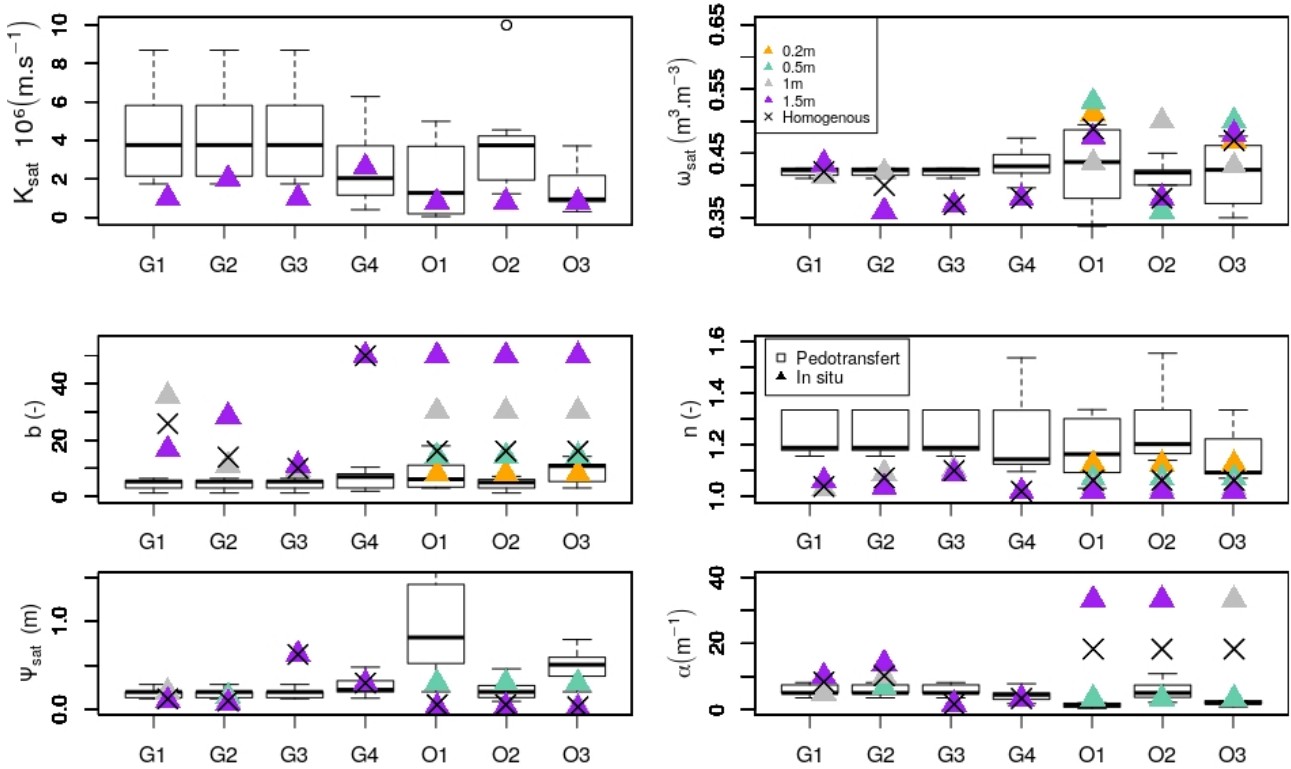

**Figure 3.** from left to right and top to bottom: Hydraulic conductivity at saturation, $k_{sat}(10^6 m.s^{-1})$, volumetric water content at saturation $\omega_{sat}(m^3.m^{-3})$, $b$ and $n$, matric potential at saturation, $\psi_{sat}(m)$, and alpha $(m)$. Estimations from in situ measurements are represented by triangles at 0.2, 0.5, 1 and 1.5 m (orange, aquamarine, grey and purple, respectively), and their mean (homogeneous) values are represented by a star. The values derived from six pedotransfer function are shown by a boxplot presenting the median, 25 % and 75 % quantiles.





**Figure 4.** Hourly time series of the total water mass variations ($kg$) around the observed or the simulated means from GISFI (G1, G2, G3, G4) and OPE (O1, O2, O3) lysimeters. Observations are in black, the BC66 model approach in red, VG80 in blue, BCVG in green, and VGc in orange. The grey shaded areas correspond to periods when the meteorological forcing is reconstructed and the blue shaded areas to the periods when data is of low quality.



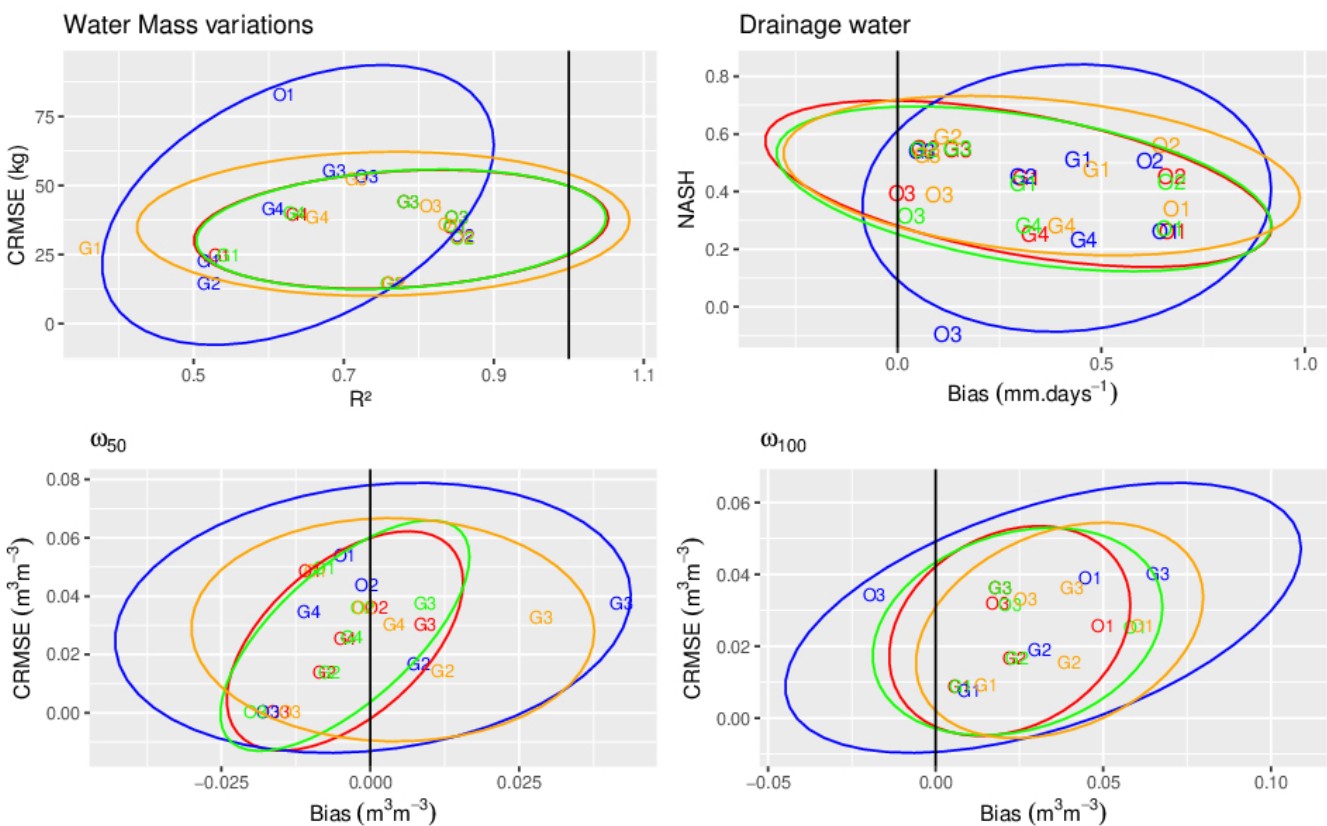

**Figure 5.** Statistical scores on daily chronicles reached by BC66 (red), VG80 (blue), BCVG (green) and VGc (orange). Top left panel shows $R^2$ *vs.* $CRMSE$ scores for the total water mass variations. Bottom left and right panels show overall Bias *vs.* $CRMSE$ scores for the volumetric water content at 0.5 m depth ($\omega_{50}$) and 1 m depth ($\omega_{100}$), respectively. Top right panel shows the overall Bias *vs.* $NASH$ scores for the drainage flux. All these skill scores are presented in Appendix A. Each lysimeter is represented by its identifier. For each model approach, ellipses represent the multivariate Student's t-distribution following Fisher (1992) .




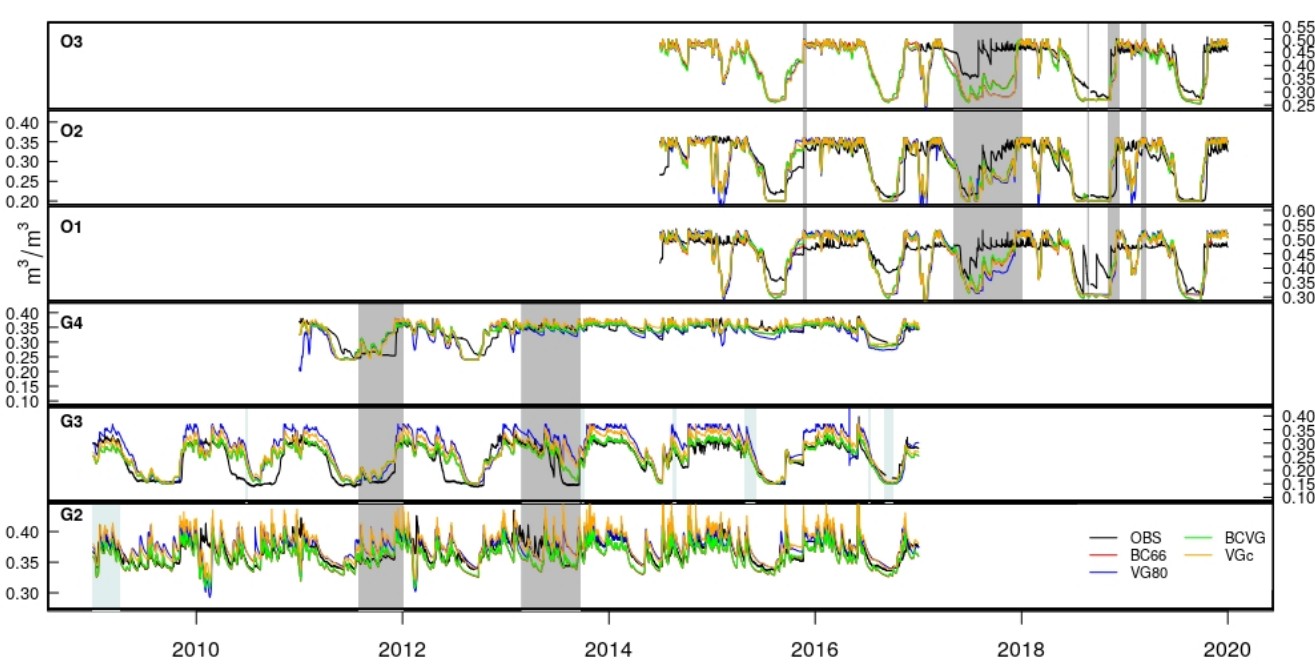

**Figure 6.** Same as figure 4 but for hourly volumetric water content (m$^3$.m$^{-3}$) at 0.5 m depth





**Figure 7.** Same as figure 4 but for daily mean annual cycles of the drainage water time series (mm.day$^{-1}$).





**Figure 8.** Same as figure 4 but for daily drainage water (mm.day$^{-1}$).

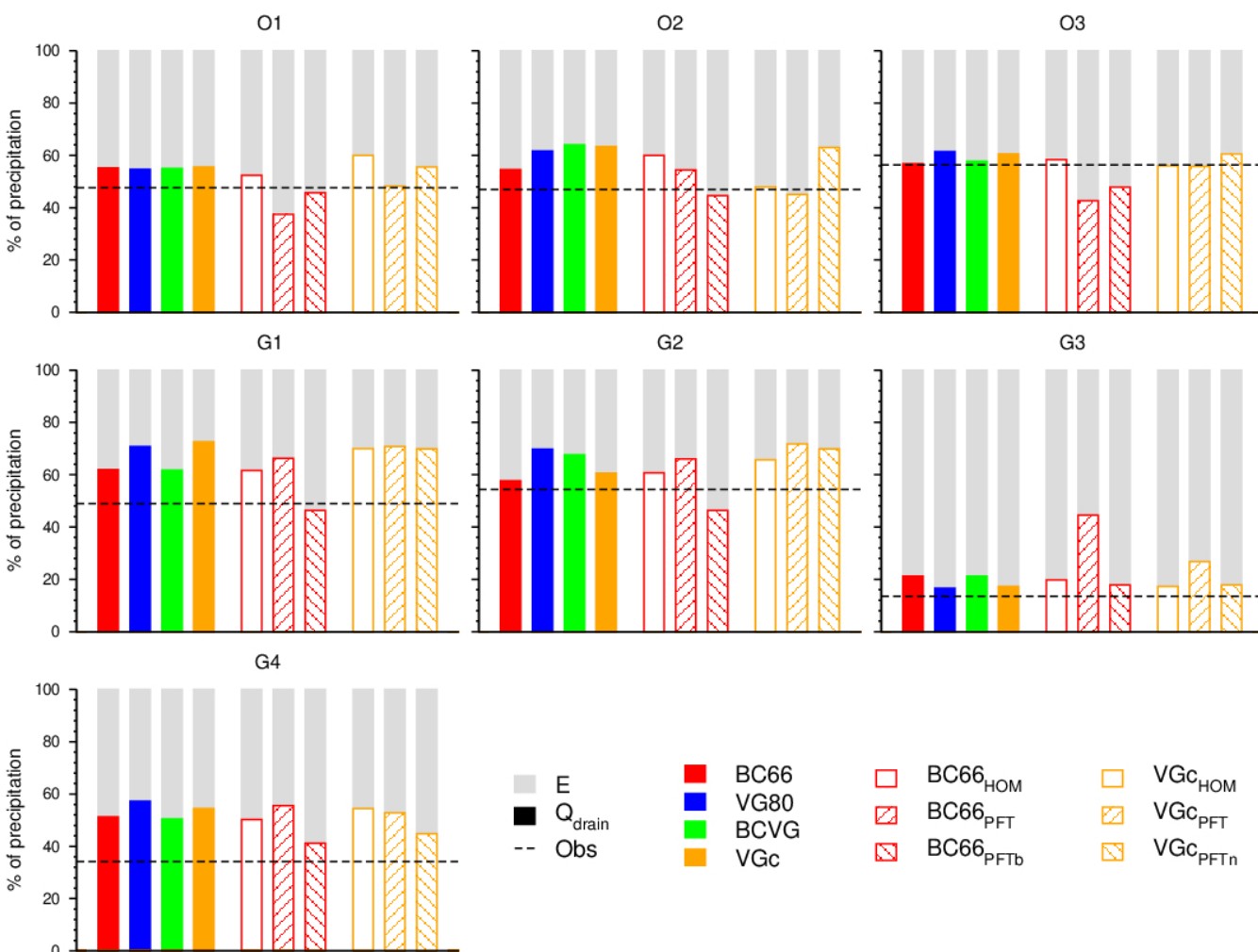

**Figure 9.** Water budget partition of the precipitation into drainage water and evapotranspiration expressed in % for lysimeters of GISFI (G1, G2, G3, G4) and OPE (O1, O2, O3) for each model approach. The observed drainage water ratio to precipitation is represented by the dashed black line. Simulated drainage water is represented by color bars for each model approach while the simulated evapotranspiration (E) is in grey. The BC66 simulated drainage water is in red, VG80 in blue, BCVG in green, and VGc in orange. Drainage water simulated by additional model approaches with an homogeneous soil profile ($BC66_{HOM}$ and $VGc_{HOM}$), with parameters estimated from usual PTFs ($BC66_{PTF}$ and $VGc_{PTF}$), and with parameters estimated with usual PTFs except $b$ ($B66_{PTFb}$) or $n$ ($VGc_{PTFn}$) estimated in situ are also shown. These last model approaches are defined in section 5.



**Figure 10.** Daily precipitation ($mm.day^{-1}$), hourly effective wetting saturation profile (%) observed (OBS) and simulated by BC66, VG80, BCVG and VGc, and daily drainage water ($mm.day^{-1}$) observed (in black), and simulated by BC66 in red, VG80 in blue, BCVG in green, and VGc in orange during intense drainage water in February 2016 for lysimeters O1 and G2. The $NASH$ scores for each simulated drainage water is also given.







**Figure 11.** Same as figure 10 but for June 2016.

**Figure 12.** Taylor diagrams for hourly Total Water Mass (a), hourly Volumetric water content at 0.5m depth (b), Daily drainage water (c) and Daily intense drainage water events (d). model approach BC66 is plotted in red, VG80 in blue, BCVG in green, and VGc in orange. Additional model approaches with an homogeneous soil profile (BCVG$_{HOM}$ and VGc$_{HOM}$) are represented as open circles, with parameters estimated from usual PTFs (BCVG$_{PTF}$ and VGc$_{PTF}$) by pluses, and with parameters estimated with PTF except $b$ (BCVG$_{PTFb}$) or $n$ (VGc$_{PTFn}$) estimated in situ by crosses. The Pearson correlation coefficient, the centred root-mean-square error ($CRMSE$), and the normalized standard deviation are summarized in this diagram (Taylor, 2001).





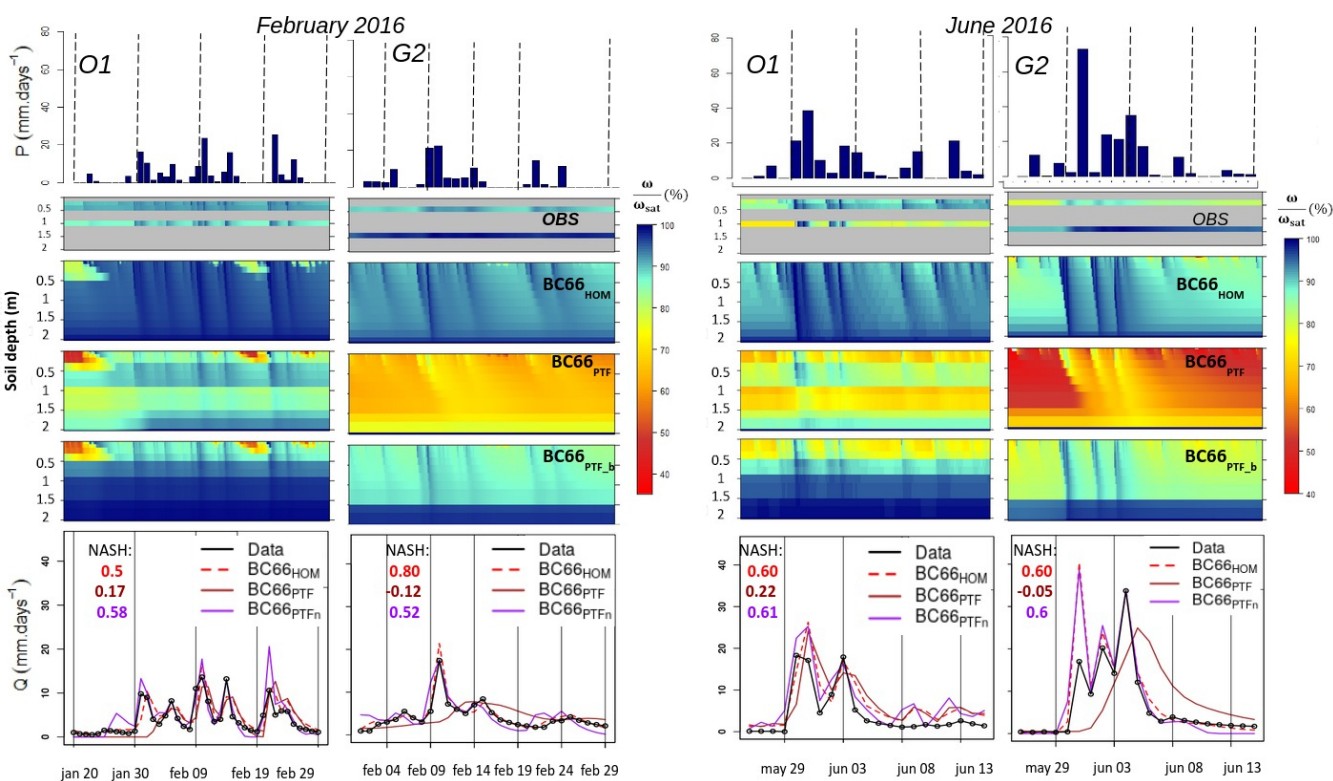

**Figure 13.** Same as Figures 10 and 11 but for sensitivity model approaches with homogeneous soil profile (BC66$_{HOM}$), with soil parameters from the usual PTFs (BC66$_{PTF}$), and with parameters from the usual PTFs except for $b$ estimated in situ (BC66$_{PTFb}$).





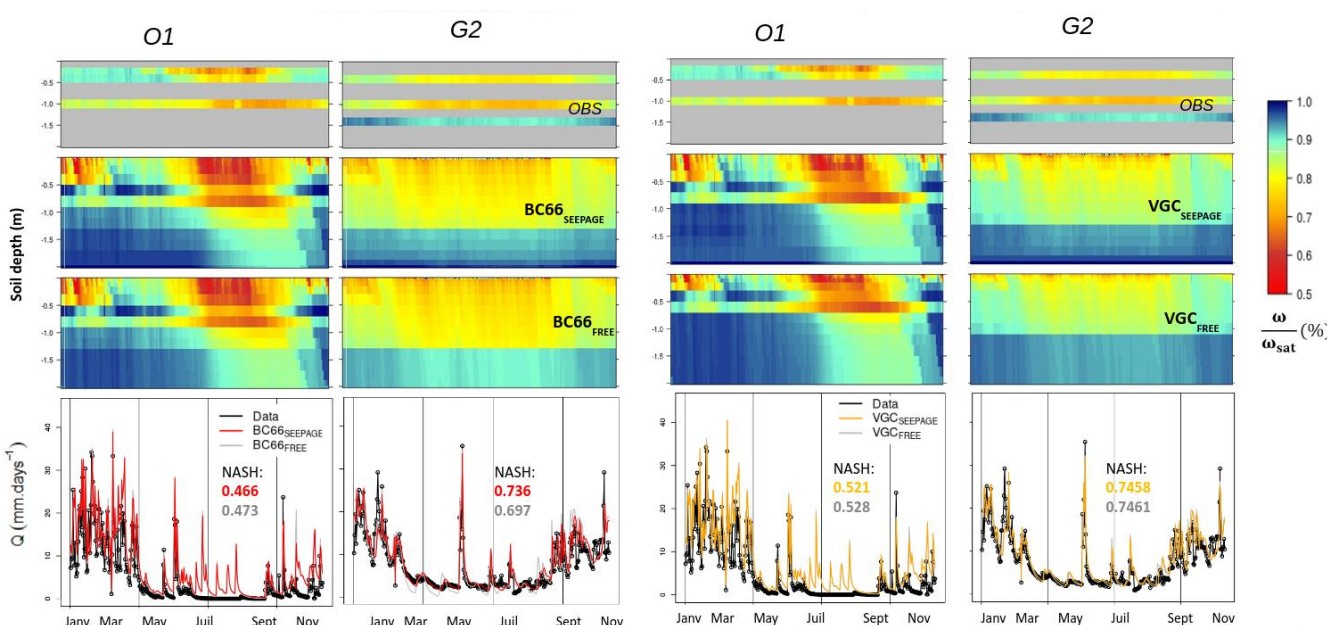

**Figure 14.** Mean annual cycles of hourly effective wetting saturation profile ($\%$) and daily drainage water ($mm.day^{-1}$) observed (OBS in black) and simulated by BC66 (left panels) and VGc (right panels) with a seepage face LBC (BC66$_{SEEPAGE}$ in red and VGc$_{SEEPAGE}$ in orange) and a free drainage LBC (BC66$_{FREE}$ and VGc$_{FREE}$ in grey) over lysimeters O1 and G2. The $NASH$ scores for each simulated drainage water is also given.