# Peer review of "Assessment of the ISBA Land Surface Model soil hydrology using four closed-form soilwater relationships and several lysimeters"

_EGUsphere, 2023_

## Referee Comment (RC1)

The paper is well written and well-illustrated. It is quite easy to read and suitable for publication in HESS. It requires some revision before publication (see comments below).

**Comments**

Richards equation is linearized with a one step Newton-Raphson scheme. How is the scheme's accuracy checked? Depending on the time step length and/or the parameter values used in the closed-form equations, the scheme accuracy is difficult to verify. Moreover, how do you handle boundary conditions that depend on the variable value like seepage?

Evaporation is assumed to take place in the first layer of one centimeter (L178). This is a very strong assumption that will limit evaporation. Please justify.

No information is provided about the intercepted rain for the lysimeters with vegetation. Is it neglected?

The artificial addition of the fine layer of 5cm (L183) is not convincingly justified. Seepage boundary conditions do not need an additional layer. To my knowledge, Tifafi et al. (2017) or Séré et al. (2012) did not use an additional layer. Please justify. Moreover, the hydraulic parameters of this layer may significantly impact drainage.

Considering hydraulic conductivity as constant over depth is a very strong assumption, especially for lysimeters G4, O1 and O3 (see soil texture Table 1). Please justify. It questions the results obtained for parameters $b$ and $n$. The key parameter is the hydraulic conductivity which is a function of hydraulic conductivity at saturation (Ksat) and the relative hydraulic conductivity that depends on $b$ and $n$. Therefore, values of $b$ and $n$ may compensate the assumption of homogeneous Ksat. You may get very similar simulations by taking $b$ and $n$ constant over depth and varying Ksat… This may also impact your conclusion (L535-536) which is 'limited' to your data and model concept. Please comment.

L342: Please define what is the 'most usable observations', is it related to number, accuracy, variability in time?

§5.1: I do not understand the interest of this paragraph. You show that the soil homogeneity assumption is not appropriate and you perform a sensitivity based on a wrong assumption. This part can be removed.

HESS Data policy. Copernicus Publications requests depositing data that correspond to journal articles in reliable (public) data repositories, assigning digital object identifiers, and properly citing data sets as individual contributions.

**Minor comments**

L174: "Crank Nicholson implicit", the time discretization is called usually Crank Nicholson, or Implicit.

L351: typo

L391-392: Are the differences really significant?

L414-415: typo

L440: I guess the mean value is the arithmetic mean? Since BC and VG are non-linear functions, the mean can be defined in several ways (like harmonic mean for hydraulic conductivity in 1D flow).

L514 : This is a numerical limitation related to your numerical model.

Séré et al, not at the right place in the references.

---

## Author Comment (AC1)

The response of the authors are noted in blue.

**Anonymous Referee 1**

The paper is well written and well-illustrated. It is quite easy to read and suitable for publication in HESS. It requires some revision before publication (see comments below).
We thank you for your general appreciation. In the following, we answer each comments you made.

**Comments**

Richards equation is linearized with a one step Newton-Raphson scheme. How is the scheme's accuracy checked? Depending on the time step length and/or the parameter values used in the closed-form equations, the scheme accuracy is difficult to verify. Moreover, how do you handle boundary conditions that depend on the variable value like seepage?
The scheme's accuracy was originally tested against a very high resolution (vertical grid) scheme using a reference fully iterative (using both the Crank-Nicolson and Backward difference in time methods) Newton Raphson scheme (Boone and Wetzel, 1996) over a range of parameter values, time steps and input infiltration rates (of varying magnitudes and forms). The linearized model was run over a range of time steps which are typically encounteref in offline and coupled model simulations (15 minutes max.) and compared this to the reference scheme.
The model is of course always stable, but indeed it was also found to be quite accurate up to time steps of around 5 minutes. Thus, if a model application uses a time step greater than this, a simple time-splitting is performed (for example, if the calling model has a time step of 5 minutes, then the Richard's routine is called thrice during this large time step. Note that for most meteorological operational applications currently, time steps on the order of seconds are used, so no splitting is required (the tendency is for smaller time steps in GCM, RCM and operational models).

Evaporation is assumed to take place in the first layer of one centimeter (L178). This is a very strong assumption that will limit evaporation. Please justify.
Here, we refer to "Bare Soil evaporation" and not to the full "Evapotranspiration" as mentioned in Mahfouf et al., 1991.
ISBA simulates the land surface evapotranspiration as the sum of the bare soil evaporation, the soil freezing sublimation, the plant transpiration, the direct evaporation of the precipitation intercepted by the plant canopy, and the snow sublimation (Noilhan and Planton, 1989). Water for bare soil evaporation is drawn from the first layer of the soil. This soil evaporation is weighted by the relative humidity of this superficial layer (Mahfouf and Noilhan, 1991). This relative humidity evolved non-linearly with the superficial water content, potentially allowing the moisture content of the soil evaporation to be greater than the usual water content at field capacity specified as matric potential at $-0.33$ bar. More detail is now added in the manuscript in the model presentation section.

No information is provided about the intercepted rain for the lysimeters with vegetation. Is it neglected?
For the lysimeters, the rain intercepted by the vegetation is assumed to evaporate or to eventually reach the ground, as it is the case for all natural surface with a vegetation canopy (now in L101).
In ISBA, the intercepted rain by vegetation is fully represented, and based on a simple rainfall interception scheme now summarized in the manuscript (L181): In ISBA, the intercepted rain by vegetation is fully represented, and based on a simple rainfall interception scheme (Noilhan and Planton, 1989; Noilhan and Mahfouf, 1996). The interception reservoir is fed by the intercepted rain by vegetation. When this reservoir is larger than its maximum value (i.e. the product of the LAI and the maximum storage of water equal to $0.2185$ kg.m$^{-2}$), the dripping from the vegetation is computed as a simple mass balance (all the water in excess drips). The direct evaporation of the vegetation is drawn from this interception reservoir and depends on the fraction of the foliage covered by intercepted water as proposed by Deardorff (1978).

The artificial addition of the fine layer of 5cm (L183) is not convincingly justified. Seepage boundary conditions do not need an additional layer. To my knowledge, Tifafi et al. (2017) or Séré et al. (2012) did not use an additional layer. Please justify. Moreover, the hydraulic parameters of this layer may significantly impact drainage.
There is probably a confusion. We talk about the discretization of the vertical domain and not to add a new layer with specific parameter. We added a fine layer (14) of 5 cm with the same parameter on the layer 13. The spatial discretization is then more finer at the bottom of the lysimeters to be more numerically stable. This approach is based on the results of Decharme et al. (2011), which demonstrated that drainage simulations were

improved with a better discretization in the Land Surface Model.

In Seré et al. (2012), authors used 2 layers with specific parameters, for a vertical domain discretized in 186 nodes.

Considering hydraulic conductivity as constant over depth is a very strong assumption, especially for lysimeters G4, O1 and O3 (see soil texture Table 1). Please justify.
It questions the results obtained for parameters b and n. The key parameter is the hydraulic conductivity, which is a function of hydraulic conductivity at saturation (Ksat) and the relative hydraulic conductivity that depends on b and n. Therefore, values of b and n may compensate the assumption of homogeneous Ksat.
You may get very similar simulations by taking b and n constant over depth and varying Ksat...
This may also impact your conclusion (L535-536) which is 'limited' to your data and model concept. Please comment.
We thank you for this interesting remark. Here we consider the hydraulic conductivity at saturation (Ksat) constant, contrary to the hydraulic conductivity (K) which evolves as a function of the water content and according to the soil profile of the others parameters.
If you speak about Ksat, of course, we agree that is a strong assumption to considering a homogeneous Ksat. On these lysimeters, we have measurements of water content and matrix pressure at several depth (20, 50, 100, 150 cm), and drainage only at 200 cm. Moreover, the unit-gradient assumption is not respected in lysimeters, and so, with these data, we cannot measure directly Ksat on the profile, and this approach is one way to access it. This approach is generally used by LSMs to estimate the Ksat.

The parameters b and n are estimated by the relation $\psi - \omega$ (Figure 1). This relation, with direct observations at different depths, give use the really values of b and n on the profile, contrary to Ksat. The relation k-$\omega$ cannot be estimated on the profile because the unit-gradient assumption is not respected in lysimeters. Only checked on the deepest measures, i.e. at 150 cm (Figure 2), this relation fitted very well with the b-n values estimated with $\psi - \omega$ and with a Ksat homogeneous.

Finally, because i) we have access to the direct value of b and n with the relation $\psi - \omega$ on the profile, and ii) we have no measurement of Ksat on the profile, it does not appear judicious to get news simulations by taking b and n constant and heterogeneous Ksat.

L342: Please define what is the 'most usable observations', is it related to number, accuracy, variability in time?
The "most usable observations" is related to number of measurement of water content. For example, in the GISFI site, there is no measurement at 20 cm. The measure are also not enough consistent at 100 and 150 cm on the OPE site, with some gaps on chronicles and inconsistent values.

§5.1: I do not understand the interest of this paragraph. You show that the soil homogeneity assumption is not appropriate and you perform a sensitivity based on a wrong assumption. This part can be removed.
In modeling at regional scales, the land surface models, as ISBA, used homogeneous soil profile. This part enable to show the importance of considering a heterogeneity profile, or not. So, we think this part is important and should not be removed.

HESS Data policy. Copernicus Publications requests depositing data that correspond to journal articles in reliable (public) data repositories, assigning digital object identifiers, and properly citing data sets as individual contributions.
The SURFEX-ISBA model is freely available here: `https://www.umr-cnrm.fr/surfex/`.
The lysimetric data are not public and are provided only on request (to be checked with the data owners: Noële Enjelvin for GISFI and Paul-Olivier Redon for OPE.) These informations are added to the article.

**Minor Comments**

L174: "Crank Nicholson implicit", the time discretization is called usually Crank Nicholson, or Implicit.
We thank you for this remark. We choose to say "Crank Nicholson".

L351: typo
We thank you for this remark.

L391-392: Are the differences really significant?
We thank you for this remark. We proposed to add this sentence : *"The absolute difference averaged over all the lysimeters is not significant different between experiments"*.

L414-415: typo
We thank you for this remark.

L440: I guess the mean value is the arithmetic mean? Since BC and VG are non-linear functions, the mean can be defined in several ways (like harmonic mean for hydraulic conductivity in 1D flow).
Yes, you are right. The mean value is computed with the arithmetic mean. We are conscious that there are several ways to consider averages on hydrological community. We choose this approach because it is the main approach used in the LSMs to homogenize soil profiles.

L514 : This is a numerical limitation related to your numerical model. Séré et al, not at the right place in the references.
We thank you for this remark.

**References :**

- Boone, A. and P. J. Wetzel, 1996: Issues related to low resolution modeling of soil moisture: Experience with the PLACE model. Glob. Plan. Change, 13, 161-181.

- Deardorff, J. W. (1978). Efficient prediction of ground surface temperature and moisture, with inclusion of a layer of vegetation. Journal of Geophysical Research: Oceans, 83(C4), 1889-1903.

- Decharme, B., Boone, A., Delire, C., Noilhan, J. (2011). Local evaluation of the Interaction between Soil Biosphere Atmosphere soil multilayer diffusion scheme using four pedotransfer functions. Journal of Geophysical Research: Atmospheres, 116(D20).

- Decharme, B., Delire, C., Minvielle, M., Colin, J., Vergnes, J. P., Alias, A., ... , Voldoire, A. (2019). Recent changes in the ISBA-CTRIP land surface system for use in the CNRM-CM6 climate model and in global off-line hydrological applications. Journal of Advances in Modeling Earth Systems, 11(5), 1207-1252.

- Mahfouf, J. F., Noilhan, J. (1991). Comparative study of various formulations of evaporation from bare soil using in situ data. Journal of Applied Meteorology (1988-2005), 1354-1365.

- Noilhan, J., Planton, S. (1989). A simple parameterization of land surface processes for meteorological models. Monthly weather review, 117(3), 536-549.

- Noilhan, J., Mahfouf, J. F. (1996). The ISBA land surface parameterisation scheme. Global and planetary Change, 13(1-4), 145-159.

- Le Moigne, P., Besson, F., Martin, E., Boé, J., Boone, A., Decharme, B., ... , Rousset-Regimbeau, F. (2020). The latest improvements with SURFEX v8. 0 of the Safran–Isba–Modcou hydrometeorological model for France. Geoscientific Model Development, 13(9), 3925-3946.

- Séré, G., Ouvrard, S., Magnenet, V., Pey, B., Morel, J. L., Schwartz, C. (2012). Predictability of the evolution of the soil structure using water flow modeling for a constructed technosol. Vadose Zone Journal, 11(1).

- Tifafi, M., Bouzouidja, R., Leguédois, S., Ouvrard, S., Séré, G. (2017). How lysimetric monitoring of Technosols can contribute to understand the temporal dynamics of the soil porosity. Geoderma, 296, 60-68.

---

## Author Comment (AC2)

The response of the authors are noted in blue.

**Anonymous referee 2**

**Comments**

The objective of the paper is to assess the ability of the $ISBA_{DF}$ land surface scheme to simulate drainage at the bottom of the soil column by comparing the simulations with data from a set of lysimeters located in two places in France, and having various soil characteristics and vegetation cover. The data are also used to compare various models for the retention and hydraulic conductivity curves used in the modeling.

I had already revised an earlier version of this paper submitted to Hydrology and Earth System Sciences. The authors have addressed the comments I had on their manuscript and I have only a few additional comments to provide.

After reading comments on the earlier version of the manuscript and those already available on the present version, I have the feeling that the authors should better specify the context of their study in the introduction. Their context is the one of Land Surface Models (LSMs) that are applied regionally or at the global scale. This context explains why numerical simulations and parameter specification are generally simplified for the model to be applicable in different contexts, without specific calibration, except in the case of some validations using in situ data at specific sites.

The present study aim is to assess the ability of the current model to simulate accurately groundwater recharge. For that, the model is applied at the local scale, using data from several lysimeter experiments. But this is not the general application context of the model, where soil parameters are computed from soil texture using PedoTransfer Functions (PTF) and the soil profile is assumed homogeneous.

The interest of the study is to show that:

- When parameters and model configuration (in particular soil vertical heterogeneity, but also lower boundary condition) is specified using in situ data, the model performance is satisfactory

- In such configuration, some combinations of soil water retention and hydraulic conductivity models provide better simulations

- Model performance is significantly decreased when vertically homogeneous soil profiles are used or when PTFs are used

The study leads to interesting conclusions with regards to the specification of soil parameters in LSMs that could be better highlighted in the abstract and the conclusions.
In view of your comments, we propose to add these comments in the text:

- Abstract (L2): *In this study, we evaluate the soil hydrology and the soil water drainage, simulated by the Interaction-Soil-Biosphere-Atmosphere (ISBA) land surface model currently used for hydrological applications from the watershed scale to the global scale, where parameters are generally not calibrated.*

- Introduction (L45): *In this context, the challenge of LSMs is to find a compromise between a simple application and an application that is powerful enough to reproduce the full water cycle. For example, in the unsaturated zone, hydrodynamic parameters are generally not calibrated and are estimated with soils properties (Decharme et al., 2011; Decharme et al., 2019; Lemoigne et al., 2020).*

- Discussion and Conclusion (L546): *In the context of LSMs that can be used at regional or global scale, the major challenge is to simplify the numerical simulations and parameter calibration, to be applicable in different contexts, without specific calibration, and to reproduce as much as possible the water cycle. This study at local scale increases the confidence that LSMs are powerful tools to simulate the recharge of groundwater, in different environmental conditions, with many soils and vegetation covers, and therefore can be used for many applications in hydrology at both the regional and the global scales.*

**Minor comments**

Line 313-314: the sentence is truncated.
We thank you for this remark.

Line 548: sensitivity instead of sensibility
We thank you for this remark.

**References :**

- Boone, A. and P. J. Wetzel, 1996: Issues related to low resolution modeling of soil moisture: Experience with the PLACE model. Glob. Plan. Change, 13, 161-181.

- Deardorff, J. W. (1978). Efficient prediction of ground surface temperature and moisture, with inclusion of a layer of vegetation. Journal of Geophysical Research: Oceans, 83(C4), 1889-1903.

- Decharme, B., Boone, A., Delire, C., Noilhan, J. (2011). Local evaluation of the Interaction between Soil Biosphere Atmosphere soil multilayer diffusion scheme using four pedotransfer functions. Journal of Geophysical Research: Atmospheres, 116(D20).

- Decharme, B., Delire, C., Minvielle, M., Colin, J., Vergnes, J. P., Alias, A., ... , Voldoire, A. (2019). Recent changes in the ISBA-CTRIP land surface system for use in the CNRM-CM6 climate model and in global off-line hydrological applications. Journal of Advances in Modeling Earth Systems, 11(5), 1207-1252.

- Mahfouf, J. F., Noilhan, J. (1991). Comparative study of various formulations of evaporation from bare soil using in situ data. Journal of Applied Meteorology (1988-2005), 1354-1365.

- Noilhan, J., Planton, S. (1989). A simple parameterization of land surface processes for meteorological models. Monthly weather review, 117(3), 536-549.

- Noilhan, J., Mahfouf, J. F. (1996). The ISBA land surface parameterisation scheme. Global and planetary Change, 13(1-4), 145-159.

- Le Moigne, P., Besson, F., Martin, E., Boé, J., Boone, A., Decharme, B., ... , Rousset-Regimbeau, F. (2020). The latest improvements with SURFEX v8. 0 of the Safran–Isba–Modcou hydrometeorological model for France. Geoscientific Model Development, 13(9), 3925-3946.

- Séré, G., Ouvrard, S., Magnenet, V., Pey, B., Morel, J. L., Schwartz, C. (2012). Predictability of the evolution of the soil structure using water flow modeling for a constructed technosol. Vadose Zone Journal, 11(1).

- Tifafi, M., Bouzouidja, R., Leguédois, S., Ouvrard, S., Séré, G. (2017). How lysimetric monitoring of Technosols can contribute to understand the temporal dynamics of the soil porosity. Geoderma, 296, 60-68.